# AQA-Bench: An Interactive Benchmark for Evaluating LLMs' Sequential Reasoning Ability in Algorithmic Environments

**Siwei Yang**[*1]    **Bingchen Zhao**[*2]    **Cihang Xie**[1]
[*]**equal technical contribution**

[1]**UC Santa Cruz**        [2]**University of Edinburgh**

**Reviewed on OpenReview:** `https://openreview.net/forum?id=W22g6Ksmbi`

## Abstract

This paper introduces AQA-Bench, a novel benchmark to assess the sequential reasoning capabilities of large language models (LLMs) in algorithmic contexts, such as depth-first search (DFS). The key feature of our evaluation benchmark lies in its interactive evaluation protocol — for example, in DFS, the availability of each node's connected edge is contingent upon the model's traversal to that node, thereby necessitating the LLM's ability to effectively remember visited nodes and strategize subsequent moves considering the possible environmental feedback in the future steps. We comprehensively build AQA-Bench with three different algorithms, namely binary search, depth-first search, and breadth-first search, and to evaluate the sequential reasoning ability of 14 different LLMs. Our investigations reveal several interesting findings: (1) Closed-source models like GPT-4 and Gemini generally show much stronger sequential reasoning ability, significantly outperforming open-source LLMs. (2) Naively providing in-context examples may inadvertently hurt few-shot performance in an interactive environment due to over-fitting to examples. (3) Instead of using optimal steps from another test case as the in-context example, a very limited number of predecessor steps in the current test case following the optimal policy can substantially boost small models' performance. (4) The performance gap between weak models and strong models is greatly due to the incapability of weak models to start well. (5) The scaling correlation between performance and model size is not always significant, sometimes even showcasing an inverse trend. We hope our study can catalyze future work on advancing the understanding and enhancement of LLMs' capabilities in sequential reasoning. The code is available at https://github.com/UCSC-VLAA/AQA-Bench.

## 1 Introduction

Recent advancements in Large Language Models (LLMs) have led to impressive strides in reasoning across a diverse array of linguistic tasks, as evidenced by a growing body of research (Wei et al., 2022; Wang et al., 2022; Brown et al., 2020; OpenAI, 2023). The reasoning capabilities of these models have typically been assessed through benchmarks focusing on arithmetic reasoning (Cobbe et al., 2021; Ling et al., 2017), symbolic inference (Suzgun et al., 2022), knowledge (Hendrycks et al., 2020), and science understanding (Hendrycks et al., 2021). These benchmarks require LLMs to engage in multi-step reasoning, leveraging both the context provided by the question and their internally learned world knowledge (Wei et al., 2022).

Nevertheless, a critical limitation of these existing benchmarks is their reliance on one-off interactions, predominantly in the form of multiple-choice questions or single-response queries. While these metrics offer valuable insights into the LLMs' reasoning abilities, they fall short in evaluating other crucial aspects of intelligence. Specifically, they do not assess the models' capacity for procedural adherence, active memory

maintenance, and ability to think ahead, which are elements vital for more complex, sequential reasoning tasks. To address this, agentic benchmarks, such as WebArena (Zhou et al., 2023), Mind2Web (Deng et al., 2023), and VisualWebArena (Koh et al., 2024), have been developed to evaluate LLMs in real-world situations where such abilities are essential. However, there are still two major drawbacks for such agentic benchmarks. First, scaling these benchmarks and quantifying environmental complexity remains difficult, complicating the systematic exploration of how these factors impact model performance. Moreover, due to the lack of known optimal strategies for these tasks, it is difficult to guide LLMs effectively and investigate the influence of various prompting techniques (Wei et al., 2022; Zhou et al.) on model outcomes.

In this work, we aim to bridge this evaluation gap in benchmarks, thereby offering a better understanding and measuring the cognitive capabilities of LLMs in mimicking human-like reasoning processes. To this end, we hereby develop an interactive Q&A benchmark, referred to as **A**lgorithmic-**QA** benchmark (AQA-Bench), specifically designed to quantitatively assess LLMs' proficiency in executing predefined algorithmic procedures. These procedures necessitate basic reasoning over observed data, coupled with the updating of an internal or external state that represents a specific data structure. One such example is solving a maze problem using the depth-first search algorithm — In each interactive instance, the model is provided with only the node ID it occupies and the edges connected to that node, representing the observed data; based on this current information and its visiting history, the model must then determine which edge to follow to progress to the subsequent node. Additionally, sequential reasoning, which is what AQA-Bench is targeting, also requires models to think ahead of the current time, considering possible environmental feedback that may receive in future steps. Take binary search as an Example. Although the probability of the targeting being any number in the given range is the same, the most optimal policy is still choosing the middle one as the current guess, as this policy can best reduce search space in the future. Through this interactive design, our AQA-Bench can effectively gauge the LLMs' capabilities in sequential reasoning in algorithmic environments. In comparison to current agentic benchmarks, our proposed AQA-Bench is more readily scalable in terms of environmental complexity and the number of test cases, facilitating alignment with the latest models. Furthermore, as the optimal strategies in our environments are known, AQA-Bench can examine the impact of various guidance forms, such as in-context examples and teacher guiding, on model performance, thus providing a more comprehensive empirical foundation for detailed analysis.

We empirically build AQA-Bench utilizing three algorithms: (1) Binary search, wherein the model's task is to deduce a number within a specified range, ideally employing the binary search algorithm. (2) Depth-first search (DFS), where the model navigates a graph to map all nodes and edges. (3) Breadth-first search (BFS), similar to DFS, but with an explicit requirement for the model to apply the BFS algorithm instead. The corresponding evaluations reveal 5 interesting findings: 1) The closed-source models like GPT-4 and Gemini strongly dominate all open-source LLMs on sequential reasoning. 2) Naively providing interactive examples may inadvertently hurt few-shot performance, probably due to overfitting with in-context learning. This trend is observed even with the advanced GPT-4 and Gemini-Pro in certain AQA-Benchenvironments. 3) Given a few predecessor steps under the optimal policy, the performance of small models can be significantly improved, sometimes even comparable to large models. 4) The difference in performance between weak and strong models largely stems from the inability of weak models to have a good initial performance. 5) The scaling correlation between performance and model size is not always signifi-

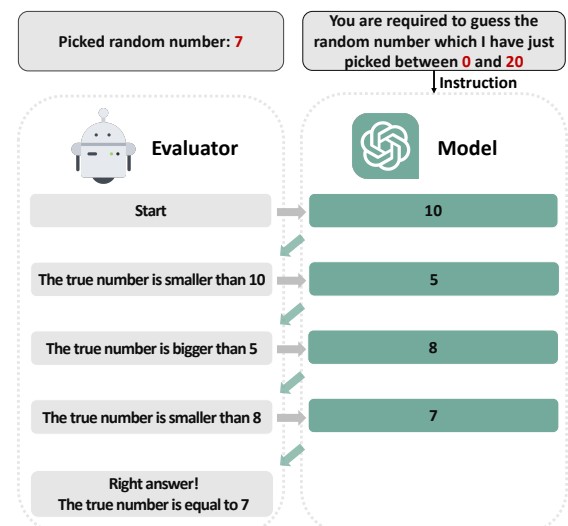

Figure 1: Illustration of AQA-Bench's evaluation process. After receiving instruction regarding the task goal, the tested model will interact with the evaluator in the form of a Q&A conversation. Metrics measuring the model's capability of achieving the task goal and following the intended algorithm are calculated based on the process. Note this is not the actual prompt we fed the model.

cant, sometimes even showcasing an inverse trend. This contradicts common assertions in LLM development and points to an oversight of sequential reasoning capabilities and "overfitting" to provided exemplars in the practice of in-context learning in current LLM research. We hope our AQA-Bench can serve as a useful benchmark for future research focused on evaluating and enhancing the sequential reasoning abilities of LLMs.

## 2  Evaluation Environments

### 2.1  Motivation behind Environment Design

AQA-Bench focuses on assessing LLMs' sequential reasoning ability with algorithmic environments mainly due to the following reasons.

**Scalable dataset.** Massive data for training and testing can be dynamically generated without human effort. It should be noted that only environment settings such as nodes and edges are pre-generated. Environments' responses to the models' interactions are not pre-defined but generated dynamically according to the models' behavior.

**Controllable complexity.** The complexity of environments can be controlled by just a few hyperparameters. Since LLMs are still evolving at a very rapid pace, traditional benchmarks may have a short life before re-development is required, while our AQA-Bench can still evaluate increasingly more powerful LLMs due to the flexible complexity of environments.

**Known optimal policy** Since the optimal policy is already known, it is much easier for us to determine whether the test model is performing the desired algorithm. This also enables teacher-guiding evaluation mode, which will be introduced in Sec. 3.3.

Additionally, this is crucial to keep the environments initially **opaque** to the tested model so that it can be forced to actively interact with the environments to gain the information necessary for performing the task via multi-time interactions. Otherwise, the task can be solved in a mega single-time generation with perhaps explicit or implicit chain-of-thought (Wei et al., 2022), and there will be no need for sequential reasoning ability.

### 2.2  Base Environment

We hereby introduce the design of three basic interactive environments. In each environment, instructions about the objective are initially fed to the model via the system prompt, while the information about the current environment is only revealed to the model following its response. Our design makes sure that the key information for making decisions can only be gained by interacting with the environment so that the model can be evaluated based on how it can plan and execute the optimal strategy. Thus, the tested model is forced to perform sequential planning by actively exploring the environment and adjusting its response according to feedback alternately.

**Base Env 1: GuessNum.** The objective of the GuessNum environment is for the model to accurately predict a number predetermined by the evaluator. During each interaction, the model interacts with the environment by guessing a number and receives feedback indicating whether its guessed number is higher or lower than the predetermined number. The optimal strategy in this scenario involves the model implementing a binary search. Consequently, the performance in this environment serves as an indicator of the model's understanding of the binary search algorithm.

**Base Env 2: DFS.** In this environment, the model is tasked with navigating a graph using the DFS algorithm. Initially, the model is presented with information about its current node and the edges connected to that node. The model interacts with the environments by deciding which edge it will follow, and then the environment will update the model with the information of the newly reached node and its associated edges. The model's performance is evaluated based on its adherence to the DFS policy. Critical to the evaluation is the ability to comprehend and implement the concept of a *first-in-last-out* stack, along with maintaining

a memory of previously visited nodes. The process of the DFS algorithm is described in the instructions to reduce difficulty.

**Base Env 3: BFS.** This environment closely mirrors the DFS environment in structure but diverges in its core algorithmic requirement, instructing the model to employ the BFS algorithm for graph navigation. This key distinction enables the BFS environment to specifically assess the model's comprehension of the *first-in-first-out* queue principle, a fundamental aspect of BFS.

## 2.3 Embodied Environment

We additionally design embodied environments where the information about each base environment is replaced with more real-life background descriptions. These embodied environments can then be used to assess if the model can perform sequential reasoning with irrelevant information, and if the model can abstract algorithmic problems from real-life situations and find the optimal algorithms.

**Embodied Env 1: Coin (GuessNum).** The tested model is required to play a hero encountering a witch guarding a chest of gold coins in a hidden temple. To claim the prize, the model needs to guess the number of gold coins with limited chances.

**Embodied Env 2: CaveDFS (DFS).** Rather than navigating a graph, the model is required to play as an explorer to visit all the caves in an underground cave system in as fewest steps possible. Unlike the DFS environment, the model is not explicitly required to use any algorithm but the objective naturally demands the DFS algorithm.

**Embodied Env 3: CaveBFS (BFS).** Similar to CaveDFS environment, this environment requests the tested model to traverse an underground cave system as well but as a group. The group can split into smaller groups to visit adjacent caves without backtracing. This environment doesn't explicitly call for a specific algorithm as well.

# 3 Evaluation

## 3.1 Metrics

To holistically assess performance in each environment, we design two specific metrics. The first is the *goal metric*, which evaluates how close is the model's final output to the ground truth; the second is the *policy metric*, which measures the efficiency of the model's policy. For the goal metric, we adopt an error-based approach where lower scores are preferable. This design choice enables the goal metric at each intermediate step can be accumulated as the policy metric to measure how fast the model's output converges to the final objective. Note that we typically prioritize the goal metric over the policy metric when comparing the performance of two models. This hierarchy in metric evaluation is crucial due to the observed tendency of lower-performing models to prematurely exit the evaluation process. Such early termination is typically a result of generating invalid responses, therefore leading to a lower goal metric score but sometimes a higher policy metric score.

**GuessNum (Coin)** requires the model to accurately guess the number specified by the evaluator. For the goal metric in this environment, we use the minimal error of the responses from the model to the target number, which is defined as

$$\text{Err}_{\min} = \max_i \frac{|g_i - \hat{g}|}{H - L + 1}, \tag{1}$$

where $g_i$ is the guess model produced in the $i$-th step of interaction, $\hat{g}$ is the target number, and $H$ and $L$ denote the upper and lower bound of the guessing range. As for the policy metric, we accumulate the error between each guess and the target number, and define the metric as:

$$\text{Err}_{\text{sum}} = \sum_i \frac{|g_i - \hat{g}|}{H - L + 1}. \tag{2}$$

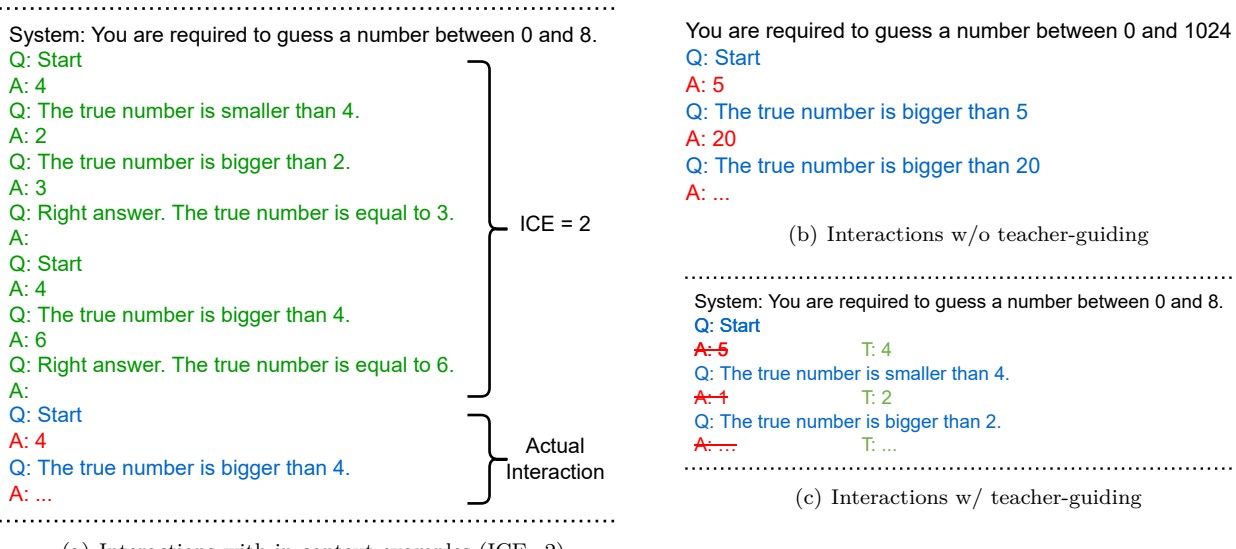

(a) Interactions with in-context examples (ICE=2)

Figure 2: **(a)** Evaluation with In-context example. The interactions of the optimal policy and the environment on the other test cases are used as examples, which can provide additional contextual information about the algorithm for in-context learning. **(b,c)** When teacher-guiding is enabled, the responses are replaced with the optimal ones to ensure no error will accumulate.

Given the similar objectives of the **DFS (CaveDFS)** and **BFS (CaveBFS)** environments, we employ a consistent metric to evaluate performance in both. The primary goal in these environments is to achieve full graph traversal. Accordingly, we define the goal metric, denoted as $G_{min}$, to measure the extent of node coverage in relation to the total number of nodes in the graph. Let $M$ represent the total number of nodes in the graph, and $\langle a \rangle_i$ denote the set of nodes visited by the model up to the $i$-th interaction step. The goal metric is then formulated as follows:

$$G_{min} = 1 - \max_i \frac{|\langle a \rangle_i|}{M} = 1 - \frac{|\langle a \rangle_{-1}|}{M}, \tag{3}$$

where $|\langle a \rangle_{-1}|$ is the number of unique nodes visited by the model by the end of the interaction.

Similarly, we define policy metric, $G_{sum}$, as the cumulative gap in graph coverage throughout the interaction:

$$G_{sum} = \sum_i 1 - \frac{|\langle a \rangle_i|}{M}.$$

Furthermore, we introduce the ratio between the step number $K_{follow}$ that the model follows the algorithm and the total number of steps the model takes $K_{total}$ to access the efficiency of models' policy:

$$ACC = \frac{K_{follow}}{K_{total}}$$

### 3.2 In-context Examples

Wei et al. (2022) argue LLM's strong reasoning abilities are, in part, attributable to their in-context learning abilities. Built upon this insight, we also incorporate it into the design of our benchmark. In Fig. 2(a), we outline our protocol for testing in-context examples within our benchmark. Specifically, it involves integrating a series of interaction examples between the optimal teacher model and the environment into the model's context. These in-context examples are expected to serve as a foundational reference, aiding the model in comprehending the expected interaction dynamics and decision-making processes in each specific environment.

### 3.3 Teacher Guiding

Directly evaluating the model on the interaction will lead to error accumulation. Such errors can result in catastrophic failure, even with strong models, due to the dependency of each step on its predecessors. However, it is also interesting to check whether correct interaction steps may also improve models' generation. To investigate this issue, we implement a strategy termed *Teacher-Guiding*. This approach involves using the intented algorithm as the optimal policy, tailored for each environment, which acts as a teacher model. The teacher model amends the outputs of the subject model, ensuring that any incorrect decision made at an intermediate step does not adversely impact subsequent interactions. The implementation of this procedure is illustrated in Fig. 2(c)[1].

We specifically designed a metric named *Per-step ACC* for this mode, defined as

$$\text{PSACC}_k = \frac{\mathcal{N}_k}{\hat{\mathcal{N}}_k} \tag{4}$$

where $\mathcal{N}_k$ is the number of test cases in which the model follows the algorithm at the $k$-th step, and $\hat{\mathcal{N}}_k$ is the number of test cases of which the optimal policy takes at least $k$ steps. Thus, $\text{PSACC}_k$ can be roughly viewed as the probability of the model following the algorithm at the $k$-th step given that the algorithm is followed in all the predecessor steps.

The averaged PSACC across all $K_{\max}$ steps in the optimal policy is used to evaluate models' overall self-guiding ability:

$$\text{PSACC}_{\text{avg}} = \frac{\sum_k \text{PSACC}_k}{K_{\max}}. \tag{5}$$

It should be noted that one major difference between teacher-guiding and in-context examples described in Sec. 3.2 is that the reference is from the current test case when evaluating with teacher-guiding while with in-context examples the reference is from other test cases.

## 4 Experiments

We evaluate models on all the environments. For the GuessNum and Coin, we set the target number between 32 and 32800, For the DFS and CaveDFS, we set the number of graph nodes to 8. For the BFS and CaveBFS, we set the number of graph nodes to 15. The worst-case runtime of the optimal policy for all environments is about 15 steps so we run evaluation with the maximum number of interactions being 20. In addition to this EASY mode, we also develop a HARD mode with a target range of $32 - 3.3 * 10^7$ for GuessNum and Coin, 13 nodes for DFS and CaveDFS, 25 nodes for BFS and CaveBFS. The optimal worst-case runtime is about 25 steps and the maximum number of interaction steps is 30. Results under the EASY mode are reported by default.

For easier comparison, we divided models into 4 categories according to the number of parameters:

- Small models with $< 10B$ parameters: Llama2-7B-chat (Touvron et al., 2023), Llama3-8B-Instruct (Team, 2024), Vicuna-7B-v1.5-16K (Chiang et al., 2023), Mistral-7B-Instruct-v0.2 (Jiang et al., 2023), DeepSeek-LLM-7B (Bi et al., 2024) and DeepSeek-MoE-16B (Dai et al., 2024).

- Medium models with $\geq 10B$ and $< 50B$ parameters: Llama2-13B-chat, Vicuna-13B-v1.5-16K, Mixtral-8x7B-Instruct-v0.1 (Jiang et al., 2024) and DeepSeek-R1-Distill-Qwen-32B (DeepSeek-AI, 2025).

- Large models with $\geq 50B$ parameters: Llama2-70B-chat and DeepSeek-LLM-67B.

- Closed-source models: GPT-3.5-Turbo, GPT-4-Turbo (OpenAI, 2023), Gemini-Pro (Team et al., 2023) and O1-Preview (OpenAI, 2024).

---

[1]Note that Figs. 2(a) and 2(c) are only for demonstration, the actual prompt fed to the model is in the Appendix D.

For mixture-of-experts models (*e.g.*, DeepSeek-MoE-16B, Mixtral-8x7B-Instruct-v0.1), we only consider the number of activated parameters during inference. All evaluations are run with zero-shot and without teacher-guiding by default.

## 4.1 Re-productivity and Variance

Although test cases in our AQA-Bench can be generated dynamically during evaluation, we still pre-generated a test set with 400 test cases for each base environment under the EASY mode and 1500 test cases under the HARD mode for simpler re-production. The final scores are averaged among test cases. A variance study on the pre-generated test set is discussed in Appendix B, showing that the evaluation results drawn from our test set can sufficiently represent the tested models' performance in environments with this level of complexity. Another factor that may affect our results is the randomness of the model itself. For open-source models and Gemini-Pro, we disable the random sampling in all the experiments. But for GPTs, which can only be accessed via OpenAI API, we cannot turn off such model randomness. However, as shown in the supplementary Appendix B.1, the variance observed in GPT models is relatively minor.

## 4.2 Main Results

**Base environments.** We start by investigating models' algorithmic sequential reasoning abilities by running evaluations in three base environments: GuessNum, DFS, and BFS. These evaluations were conducted naively, without the incorporation of in-context examples or teacher guidance. As shown in Tab. 1, closed-source models like GPTs and Gemini generally exhibit much superior performance compared to most of the tested open-source models; The only exception among tested environments is the DFS environment, where open-source models outperform GPT-3.5-Turbo, but are still not as good as GPT-4-Turbo and Gemini-Pro.

Next, among open-source models, one interesting observation is that more recently released models (*e.g.*, Mistral, Deepseek-LLM) are arguably better than relatively older ones (*e.g.*, Llama, Vicuna). For example, Mixtral-8x7B-Instruct-v0.1, which is claimed to be better than Llama2-70B-chat, does excel Llama2-70B-chat in GuessNum and BFS but falls short in DFS. As for the DeepSeek-MoE-16B model, which out-performs Llama2-7B-chat on conventional language benchmarks (Bi et al., 2024), underperforms Llama2-7B-chat across all three tested environments. Notably, Llama3-70B-Instruct, being the best open-source non-reasoning we tested, still manages to achieve higher performance than GPT-3.5-Turbo, even surpass closed-source models in CaveBFS.

As for the recently trending reasoning models, it is particularly worth mentioning that O1-Preview almost achieves the task goal in all test cases with a substantially low goal metric and much higher accuracy, suggesting that strong reasoning models can not only excel in single-round Q&A tasks but also in interactive environments that require long-term planning.

The other reasoning model we tested is DeepSeek-R1-Distill-Qwen-32B, showing much better ACCs but worse goal metrics than other medium models, especially under HARD Mode, as shown in Tab. 1. This suggests that for a weaker reasoning model, it is better at following the optimal policy at the beginning of a full trajeotory but quickly failed to adhere, resulting in an early termination of the interaction and much worse goal metrics, which is likely due to the increased token length of reasoning models. Unlike O1-Preview, a reasoning model built upon a weaker base model with a much shorted supported token length might be greatly held back by this issue, which is a lot more server in interactive environments compared to single-round Q&A tasks.

**Embodied environments.** The findings from the embodied environments, as detailed in Tab. 1, largely mirror the conclusions drawn from the base environments. Moreover, as shown in Fig. 3, we interestingly note that models don't consistently perform worse in embodied environments, although these embodied environments require models to implicitly abstract from the environments and decide the optimal algorithm to execute, thus are expected to be more difficult for models.

Table 1: **The main evaluation results with all environments.** For models with strong goal metrics (*e.g.*, $\text{Err}_{\min}$, $\text{G}_{\min}$) indicting weak performance, goal metrics are more informative than policy metrics (*e.g.*, $\text{Err}_{\text{sum}}$, $\text{G}_{\text{sum}}$, ACC).

**Base Envs under EASY Mode**

| Model | GuessNum | | DFS | | BFS | |
|---|---|---|---|---|---|---|
| | $\text{Err}_{\min}\downarrow$ | ACC↑ | $\text{G}_{\min}\downarrow$ | ACC↑ | $\text{G}_{\min}\downarrow$ | ACC↑ |
| Small < 10B | | | | | | |
| Llama2-7B-chat | 0.26 | 0.00 | 0.58 | 0.24 | 0.60 | 0.00 |
| Llama3-8B-Instruct | 0.01 | 0.00 | 0.21 | 0.51 | 0.02 | 0.23 |
| Vicuna-7B | 0.46 | 0.00 | 0.65 | 0.15 | 0.84 | 0.03 |
| Mistral-7B-Instruct-v0.2 | 0.06 | 0.00 | 0.49 | 0.61 | 0.24 | 0.13 |
| DeepSeek-LLM-7B | 0.43 | 0.00 | 0.34 | 0.36 | 0.52 | 0.06 |
| DeepSeek-MoE-16B | 1.00 | 0.00 | 0.63 | 0.07 | 0.88 | 0.02 |
| 10B ≤ Medium < 50B | | | | | | |
| Llama2-13B-chat | 0.01 | 0.00 | 0.34 | 0.41 | 0.65 | 0.05 |
| Vicuna-13B | 0.39 | 0.00 | 0.66 | 0.12 | 0.81 | 0.05 |
| Mixtral-8x7B-Instruct-v0.1 | 0.00 | 0.00 | 0.47 | 0.57 | 0.14 | 0.21 |
| DeepSeek-R1-Distill-Qwen-32B | 0.12 | 0.05 | 0.58 | 0.43 | 0.72 | 0.13 |
| Large ≥ 50B | | | | | | |
| Llama2-70B-chat | 0.11 | 0.00 | 0.33 | 0.44 | 0.28 | 0.06 |
| Llama3-70B-Instruct | 0.01 | 0.00 | 0.10 | 0.68 | 0.01 | **0.53** |
| DeepSeek-LLM-67B | 0.12 | 0.00 | 0.40 | 0.42 | 0.45 | 0.09 |
| Closed-source | | | | | | |
| GPT-3.5-Turbo | 0.00 | 0.01 | 0.35 | 0.61 | 0.11 | 0.52 |
| GPT-4-Turbo | **0.00** | **0.46** | 0.03 | 0.94 | **0.00** | 0.38 |
| Gemini-Pro | 0.00 | 0.00 | 0.25 | 0.76 | 0.06 | 0.17 |
| O1-Preview | **0.00** | 0.43 | **0.00** | **1.00** | **0.00** | **0.99** |

**Embodied Envs under EASY Mode**

| Model | Coin | | CaveDFS | | CaveBFS | |
|---|---|---|---|---|---|---|
| | $\text{Err}_{\min}\downarrow$ | ACC↑ | $\text{G}_{\min}\downarrow$ | ACC↑ | $\text{G}_{\min}\downarrow$ | ACC↑ |
| Small < 10B | | | | | | |
| Llama2-7B-chat | 0.07 | 0.00 | 0.50 | 0.33 | 0.76 | 0.05 |
| Llama3-8B-Instruct | 0.07 | 0.00 | 0.17 | 0.40 | 0.10 | 0.16 |
| Vicuna-7B | 1.00 | 0.00 | 0.54 | 0.21 | 0.72 | 0.07 |
| Mistral-7B-Instruct-v0.2 | 0.07 | 0.00 | 0.49 | 0.48 | 0.27 | 0.11 |
| DeepSeek-LLM-7B | 0.39 | 0.00 | 0.58 | 0.16 | 0.77 | 0.04 |
| DeepSeek-MoE-16B | 1.00 | 0.00 | 0.71 | 0.11 | 0.89 | 0.01 |
| 10B ≤ Medium < 50B | | | | | | |
| Llama2-13B-chat | 0.19 | 0.00 | 0.38 | 0.36 | 0.55 | 0.09 |
| Vicuna-13B | 1.00 | 0.00 | 0.56 | 0.21 | 0.64 | 0.06 |
| Mixtral-8x7B-Instruct-v0.1 | 0.00 | 0.00 | 0.32 | 0.45 | 0.15 | **0.17** |
| DeepSeek-R1-Distill-Qwen-32B | 0.18 | 0.06 | 0.72 | 0.13 | 0.58 | 0.44 |
| Large ≥ 50B | | | | | | |
| Llama2-70B-chat | **0.00** | 0.00 | 0.35 | 0.44 | 0.30 | 0.03 |
| Llama3-70B-Instruct | 0.03 | 0.00 | 0.03 | 0.76 | **0.01** | **0.17** |
| DeepSeek-LLM-67B | 0.36 | 0.00 | 0.28 | 0.57 | 0.38 | 0.08 |
| Closed-source | | | | | | |
| GPT-3.5-Turbo | **0.00** | 0.00 | 0.20 | 0.66 | 0.27 | 0.10 |
| GPT-4-Turbo | **0.00** | **0.50** | 0.23 | 0.74 | 0.12 | 0.16 |
| Gemini-Pro | **0.00** | 0.00 | 0.22 | 0.70 | 0.10 | 0.16 |
| O1-Preview | **0.00** | 0.05 | **0.00** | **1.00** | 0.12 | 0.08 |

**Base Envs under HARD Mode**

| Model | GuessNum | | DFS | | BFS | |
|---|---|---|---|---|---|---|
| | $\text{Err}_{\min}\downarrow$ | ACC↑ | $\text{G}_{\min}\downarrow$ | ACC↑ | $\text{G}_{\min}\downarrow$ | ACC↑ |
| Small < 10B | | | | | | |
| Llama2-7B-chat | 0.49 | 0.00 | 0.74 | 0.19 | 0.76 | 0.01 |
| Llama3-8B-Instruct | 0.07 | 0.00 | 0.41 | 0.43 | 0.07 | 0.13 |
| Vicuna-7B | 0.24 | 0.00 | 0.78 | 0.10 | 0.89 | 0.02 |
| Mistral-7B-Instruct-v0.2 | 0.06 | 0.00 | 0.65 | 0.61 | 0.46 | 0.08 |
| DeepSeek-LLM-7B | 0.49 | 0.00 | 0.61 | 0.18 | 0.71 | 0.04 |
| DeepSeek-MoE-16B | 1.00 | 0.00 | 0.78 | 0.03 | 0.92 | 0.01 |
| 10B ≤ Medium < 50B | | | | | | |
| Llama2-13B-chat | 0.49 | 0.00 | 0.59 | 0.25 | 0.76 | 0.03 |
| Vicuna-13B | 0.49 | 0.00 | 0.80 | 0.07 | 0.83 | 0.03 |
| Mixtral-8x7B-Instruct-v0.1 | 0.00 | 0.00 | 0.64 | 0.58 | 0.32 | 0.13 |
| DeepSeek-LLM-67B | **0.00** | 0.00 | 0.51 | 0.28 | 0.67 | 0.05 |
| Large ≥ 50B | | | | | | |
| Llama2-70B-chat | 0.49 | 0.00 | 0.48 | 0.35 | 0.43 | 0.04 |
| Llama3-70B-Instruct | **0.00** | 0.00 | 0.25 | 0.56 | 0.02 | 0.37 |
| DeepSeek-LLM-67B | **0.00** | 0.00 | 0.51 | 0.28 | 0.67 | 0.05 |
| Closed-source | | | | | | |
| GPT-3.5-Turbo | **0.00** | 0.00 | 0.55 | 0.51 | 0.27 | 0.29 |
| GPT-4-Turbo | **0.00** | 0.04 | 0.08 | 0.87 | 0.01 | 0.26 |
| Gemini-Pro | **0.00** | 0.00 | 0.33 | 0.69 | 0.12 | 0.09 |
| O1-Preview | **0.00** | **0.22** | **0.00** | **0.99** | **0.00** | **0.96** |

**Embodied Envs under HARD Mode**

| Model | Coin | | CaveDFS | | CaveBFS | |
|---|---|---|---|---|---|---|
| | $\text{Err}_{\min}\downarrow$ | ACC↑ | $\text{G}_{\min}\downarrow$ | ACC↑ | $\text{G}_{\min}\downarrow$ | ACC↑ |
| Small < 10B | | | | | | |
| Llama2-7B-chat | 0.49 | 0.00 | 0.68 | 0.19 | 0.83 | 0.04 |
| Llama3-8B-Instruct | 0.07 | 0.00 | 0.29 | 0.29 | 0.22 | 0.09 |
| Vicuna-7B | 0.49 | 0.00 | 0.70 | 0.13 | 0.83 | 0.05 |
| Mistral-7B-Instruct-v0.2 | 0.08 | 0.00 | 0.61 | 0.50 | 0.49 | 0.07 |
| DeepSeek-LLM-7B | 0.49 | 0.00 | 0.74 | 0.11 | 0.86 | 0.03 |
| DeepSeek-MoE-16B | 1.00 | 0.00 | 0.86 | 0.05 | 0.94 | 0.01 |
| 10B ≤ Medium < 50B | | | | | | |
| Llama2-13B-chat | 0.08 | 0.00 | 0.56 | 0.28 | 0.68 | 0.06 |
| Vicuna-13B | 1.00 | 0.00 | 0.65 | 0.17 | 0.71 | 0.05 |
| Mixtral-8x7B-Instruct-v0.1 | 0.07 | 0.00 | 0.50 | 0.38 | 0.30 | 0.09 |
| DeepSeek-R1-Distill-Qwen-32B | 0.19 | 0.07 | 0.73 | 0.38 | 0.84 | 0.13 |
| Large ≥ 50B | | | | | | |
| Llama2-70B-chat | 0.08 | 0.00 | 0.49 | 0.33 | 0.46 | 0.02 |
| Llama3-70B-Instruct | **0.00** | 0.00 | **0.13** | 0.56 | **0.05** | **0.09** |
| DeepSeek-LLM-67B | 0.02 | 0.00 | 0.39 | 0.40 | 0.56 | 0.06 |
| Closed-source | | | | | | |
| GPT-3.5-Turbo | 0.37 | 0.00 | 0.33 | 0.56 | 0.45 | 0.07 |
| GPT-4-Turbo | **0.00** | 0.04 | 0.33 | 0.67 | 0.19 | **0.09** |
| Gemini-Pro | **0.00** | 0.00 | 0.35 | 0.56 | 0.23 | **0.09** |
| O1-Preview | **0.00** | **0.22** | **0.00** | **0.99** | 0.22 | 0.05 |

## 4.3 Effect of In-Context Examples

This section explores the impact of introducing in-context examples on different models. The results, as detailed in Tabs. 2 and 3, showcase that most models get significant improvement when provided with in-context examples. For example, initially, in the absence of in-context examples (ICE=0), DeepSeek-MoE-7B is outperformed by Llama2-7B-chat across all six environments; but when presented with more in-context examples, DeepSeek-MoE-7B not only bridges the performance gap but actually surpasses Llama2-7B-chat in effectiveness.

However, the benefit of in-context examples is not universally observed across all models. For instance, the Llama2-13B-chat model exhibits a decline in performance in the DFS environment when presented with seven

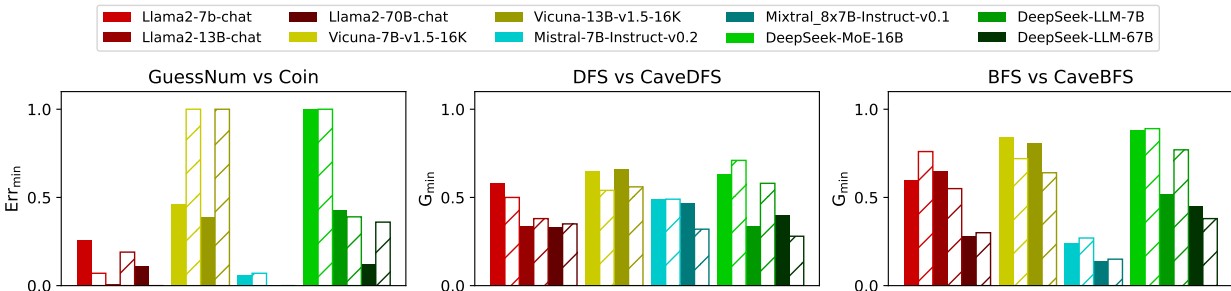

Figure 3: Goal metrics from all 6 environments. Bars that are fully filled represent results from base environments (*e.g.*, GuessNum, DFS, BFS) while bars filled with diagonal lines are from embodied environments (*e.g.*, Coin, CaveDFS, CaveBFS). Models in the same family are represented with the same hue, and larger models correspond to darker colors. Results from GPTs and Gemini-Pro are not shown in this figure. It should be noted that these metrics are the lower the better.

Table 2: The evaluation results (ICE=7) in 3 base environments. For models with strong goal metrics (*e.g.*, $Err_{min}$, $G_{min}$) indicting weak performance, goal metrics are more informative than ACC.

| Model | GuessNum | | DFS | | BFS | |
|---|---|---|---|---|---|---|
| | $Err_{min} \downarrow$ | ACC $\uparrow$ | $G_{min} \downarrow$ | ACC $\uparrow$ | $G_{min} \downarrow$ | ACC $\uparrow$ |
| Small < 10B | | | | | | |
| Llama2-7B-chat | 0.08 (-0.18) | 0.10 (+0.10) | 0.39 (-0.19) | 0.23 (-0.01) | 0.65 (+0.05) | 0.14 (+0.14) |
| Llama3-8B-Instruct | 0.02 (+0.01) | 0.19 (+0.19) | 0.04 (-0.17) | 0.48 (-0.03) | 0.37 (+0.35) | 0.12 (-0.11) |
| Vicuna-7B | 0.02 (-0.44) | 0.22 (+0.22) | 0.37 (-0.28) | 0.27 (+0.12) | 0.68 (-0.16) | 0.15 (+0.12) |
| Mistral-7B-Instruct-v0.2 | 0.01 (-0.05) | 0.22 (+0.22) | 0.14 (-0.35) | 0.51 (-0.10) | 0.39 (+0.15) | 0.17 (+0.04) |
| DeepSeek-LLM-7B | 0.04 (-0.39) | 0.18 (+0.18) | 0.16 (-0.18) | 0.17 (-0.19) | 0.61 (+0.09) | 0.18 (+0.12) |
| DeepSeek-MoE-16B | 0.02 (-0.98) | 0.21 (+0.21) | 0.14 (-0.49) | 0.30 (+0.23) | 0.86 (-0.02) | 0.10 (+0.08) |
| 10B ≤ Medium < 50B | | | | | | |
| Llama2-13B-chat | 0.06 (+0.05) | 0.13 (+0.13) | 0.50 (+0.16) | 0.18 (-0.23) | 0.57 (-0.08) | 0.09 (+0.04) |
| Vicuna-13B | 0.12 (-0.27) | 0.12 (+0.12) | 0.16 (-0.50) | 0.63 (+0.51) | 0.23 (-0.58) | 0.27 (+0.22) |
| Mixtral-8x7B-Instruct-v0.1 | 0.00 (+0.00) | 0.25 (+0.25) | 0.20 (-0.27) | 0.44 (-0.13) | 0.48 (+0.34) | 0.21 (+0.00) |
| Large ≥ 50B | | | | | | |
| Llama2-70B-chat | 0.07 (-0.04) | 0.13 (+0.13) | 0.14 (-0.19) | 0.46 (+0.02) | 0.46 (+0.18) | 0.11 (+0.05) |
| Llama3-70B-Instruct | 0.01 (+0.00) | 0.19 (+0.19) | **0.00** (-0.10) | 0.72 (+0.04) | **0.00** (-0.01) | 0.37 (-0.16) |
| DeepSeek-LLM-67B | **0.00** (-0.12) | 0.25 (+0.25) | 0.18 (-0.22) | 0.33 (-0.09) | 0.36 (-0.09) | 0.18 (+0.09) |
| Closed-source | | | | | | |
| GPT-3.5-Turbo | **0.00** (+0.00) | 0.01 (+0.00) | 0.36 (+0.01) | 0.62 (+0.01) | 0.12 (+0.01) | **0.51** (-0.01) |
| GPT-4-Turbo | **0.00** (+0.00) | **0.47** (+0.01) | 0.02 (-0.01) | **0.94** (+0.00) | **0.00** (+0.00) | 0.40 (+0.02) |
| Gemini-Pro | **0.00** (+0.00) | 0.43 (+0.43) | 0.02 (-0.23) | 0.68 (-0.08) | 0.03 (-0.03) | 0.36 (+0.19) |

in-context examples (ICE=7). To delve deeper into this phenomenon, we analyze the performance variation in relation to the number of in-context examples, as depicted in Fig. 4. Two interesting observations are noted: 1) For GPT models, in-context learning barely had any impact on their performance, even though there is still room for improvement in embodied environments; and 2) An intriguing pattern emerged among the Llama2 models in the Coin environment, where their performance significantly dropped with just one in-context example (ICE=1), but showed gradual improvement as the number of examples increased. Similar trends were observed in recent open-source models, such as Mistral-7B in BFS, DeepSeek-67B in Coin and closed-source Gemini-Pro in BFS. This contradicts the typical assumption that in-context learning universally enhances LLMs' performance. We hypothesize that this contradiction may stem from the interactive and

Table 3: The evaluation results (ICE=7) in 3 embodied environments. For models with strong goal metrics (*e.g.*, $\text{Err}_{min}$, $\text{G}_{min}$) indicting weak performance, goal metrics are more informative than policy metrics (*e.g.*, $\text{Err}_{sum}$, $\text{G}_{sum}$, ACC).

| Model | Coin | | CaveDFS | | CaveBFS | |
|---|---|---|---|---|---|---|
| | $\text{Err}_{min} \downarrow$ | ACC $\uparrow$ | $\text{G}_{min} \downarrow$ | ACC $\uparrow$ | $\text{G}_{min} \downarrow$ | ACC $\uparrow$ |
| Small < 10B | | | | | | |
| Llama2-7B-chat | 0.11 (+0.04) | 0.08 (+0.08) | 0.38 (-0.12) | 0.26 (-0.07) | 0.58 (-0.18) | 0.11 (+0.06) |
| Llama3-8B-Instruct | 0.02 (-0.05) | 0.19 (+0.19) | 0.04 (-0.13) | 0.49 (+0.09) | 0.42 (+0.32) | 0.10 (-0.06) |
| Vicuna-7B | 0.02 (-0.98) | 0.22 (+0.22) | 0.39 (-0.15) | 0.25 (+0.04) | 0.68 (-0.04) | 0.14 (+0.07) |
| Mistral-7B-Instruct-v0.2 | 0.01 (-0.06) | 0.22 (+0.22) | 0.18 (-0.31) | 0.45 (-0.03) | 0.48 (+0.21) | 0.15 (+0.04) |
| DeepSeek-LLM-7B | 0.04 (-0.35) | 0.17 (+0.17) | 0.19 (-0.39) | 0.33 (+0.17) | 0.62 (-0.15) | 0.17 (+0.13) |
| DeepSeek-MoE-16B | 0.02 (-0.98) | 0.21 (+0.21) | 0.13 (-0.58) | 0.34 (+0.23) | 0.87 (-0.02) | 0.08 (+0.07) |
| $10B \leq$ Medium $< 50B$ | | | | | | |
| Llama2-13B-chat | 0.05 (-0.14) | 0.13 (+0.13) | 0.48 (+0.10) | 0.19 (-0.17) | 0.56 (+0.01) | 0.11 (+0.02) |
| Vicuna-13B | 0.13 (-0.87) | 0.12 (+0.12) | 0.15 (-0.41) | 0.59 (+0.38) | 0.27 (-0.37) | 0.27 (+0.21) |
| Mixtral-8x7B-Instruct-v0.1 | 0.00 (+0.00) | 0.23 (+0.23) | 0.17 (-0.15) | 0.43 (-0.02) | 0.39 (+0.24) | 0.21 (+0.04) |
| Large $\geq 50B$ | | | | | | |
| Llama2-70B-chat | 0.09 (+0.09) | 0.12 (+0.12) | 0.20 (-0.15) | 0.42 (-0.02) | 0.60 (+0.30) | 0.08 (+0.05) |
| Llama3-70B-Instruct | 0.01 (-0.02) | 0.17 (+0.17) | **0.00** (-0.03) | 0.75 (-0.01) | **0.01** (+0.00) | 0.32 (+0.15) |
| DeepSeek-LLM-67B | **0.00** (-0.36) | 0.24 (+0.24) | 0.18 (-0.10) | 0.37 (-0.20) | 0.39 (+0.01) | 0.21 (+0.13) |
| Closed-source | | | | | | |
| GPT-3.5-Turbo | 0.02 (+0.02) | 0.00 (+0.00) | 0.19 (-0.01) | 0.66 (+0.00) | 0.27 (+0.00) | 0.10 (+0.00) |
| GPT-4-Turbo | **0.00** (+0.00) | **0.50** (+0.00) | 0.23 (+0.00) | **0.76** (+0.02) | 0.11 (-0.01) | 0.16 (+0.00) |
| Gemini-Pro | **0.00** (+0.00) | 0.41 (+0.41) | 0.04 (-0.18) | 0.54 (-0.16) | 0.05 (-0.05) | **0.33** (+0.17) |

Figure 4: Goal metrics from all 6 environments. It should be noted that these metrics are the lower the better. Large models are more prune to performance drop when ICE is provided.

multi-round nature of examples in AQA-Bench, as opposed to the single-round format typical in standard Q&A benchmarks.

This suggests that more studies about how multi-round examples for interactive tasks should be given to LLMs are required. It is also worth noting that with ICE=7, Gemini-Pro showed comparable or even better performance than GPT-4-Turbo in all environments except CaveDFS. Lastly, we investigate the influence of instructional differences between the base environments and their embodied variants, particularly with the increasing number of in-context examples. In Fig. 4, we observe a notable trend: as the number of

in-context examples increases, the disparity in goal metrics between most models across these two types of environments tends to diminish. This suggests that the method of instruction and example provision can substantially reshape model behaviors.

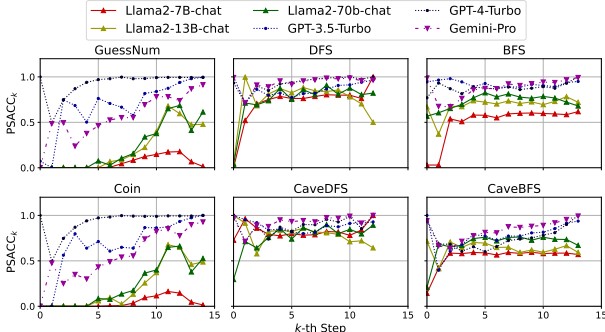

Figure 5: Per-step ACC in all 6 environments. Weak models are greatly improved even when a few optimal predecessor steps are provided.

Table 4: $\text{PSACC}_{\text{avg}}$ in all 6 environments. All tests are run w/ teacher-guiding.

| Model | GuessNum ↑ | DFS ↑ | BFS ↑ | Coin ↑ | CaveDFS ↑ | CaveBFS ↑ |
|---|---|---|---|---|---|---|
| | Small < 10B | | | | | |
| Llama2-7B-chat | 0.06 | 0.71 | 0.50 | 0.04 | 0.83 | 0.54 |
| Llama3-8B-Instruct | 0.26 | 0.76 | 0.85 | 0.25 | 0.77 | 0.80 |
| Vicuna-7B | 0.04 | 0.80 | 0.57 | 0.03 | 0.82 | 0.61 |
| Mistral-7B-Instruct-v02 | 0.09 | 0.77 | 0.74 | 0.09 | 0.75 | 0.74 |
| DeepSeek-LLM-7B | 0.02 | 0.81 | 0.60 | 0.01 | 0.77 | 0.60 |
| DeepSeek-MOE-16B | 0.04 | 0.68 | 0.60 | 0.05 | 0.71 | 0.28 |
| | $10B \leq$ Medium $< 50B$ | | | | | |
| Llama2-13B-chat | 0.21 | 0.74 | 0.69 | 0.21 | 0.79 | 0.63 |
| Vicuna-13B | 0.19 | 0.77 | 0.72 | 0.18 | 0.82 | 0.75 |
| Mixtral-8x7B-Instruct-v01 | 0.44 | 0.79 | 0.86 | 0.45 | 0.85 | 0.81 |
| | Large $\geq 50B$ | | | | | |
| Llama2-70B-chat | 0.23 | 0.75 | 0.73 | 0.23 | 0.76 | 0.68 |
| Llama3-70B-Instruct | 0.56 | 0.76 | 0.91 | 0.46 | 0.72 | 0.83 |
| DeepSeek-LLM-67B | 0.35 | 0.87 | 0.69 | 0.41 | 0.91 | 0.68 |
| | Closed-source | | | | | |
| GPT-3.5-Turbo | 0.68 | 0.86 | **0.92** | 0.67 | 0.89 | 0.77 |
| GPT-4-Turbo | **0.93** | 0.93 | 0.88 | **0.93** | 0.89 | 0.75 |
| Gemini-Pro | 0.56 | **0.94** | 0.88 | 0.56 | **0.93** | **0.84** |

## 4.4 Failure of Scaling Law

**Zero-shot setting.** Fig. 3 provides comparisons among models from the same family. An interesting observation is, contrary to the expected improvement with increased model size — a trend typically observed in LLM benchmarks — the performance in tasks like GuessNum, DFS, and their embodied variants does not consistently correlate with larger model sizes. Notably, certain models exhibit an inverse scaling effect. For instance, DeepSeek-LLM-7B surpasses its larger 65B counterpart in the DFS environment. Similarly, Llama-7B-chat outperforms the 13B one in the Coin environment.

**Few-shot setting.** The deviation from the scaling law becomes even more pronounced in the few-shot settings, as evidenced in Fig. 4. In these scenarios, medium and large models more frequently experience performance drops compared to smaller models. This phenomenon is probably caused during the overfitting process from in-context learning of medium and large models, which is also an overlooked area. This pattern also suggests that while larger models are often touted by developers for their superior performance across a range of benchmarks, their effectiveness may not uniformly extend to specialized domains such as algorithmic execution and interactive sequential reasoning. In these areas, the challenges are distinct from those encountered in conventional one-round Q&A formats, indicating a need to reconsider the scaling assumptions in LLM for these applications.

## 4.5 Teacher Guiding

As evidenced in Fig. 5, even Llama2-7B-chat, which is a small model, yielded higher PSACC as the number of steps grows, indicating the probability of executing the optimal policy improves over time, especially as correct decisions accumulate. In environments like DFS (CaveDFS) and BFS (CaveBFS), we noted that the differences in PSACC among various Llama2 models diminish when more guidance steps are provided by the teacher model. While larger models still tend to exhibit a higher average PSACC, as shown in Tab. 4, the gap narrows with increased teacher model intervention. However, it is important to note, as in Fig. 5, PSACC may begin to decline in later stages of interaction. This decline can be attributed to the complexity of adhering to the optimal policy as the model is required to track and remember previous steps, such as (implicitly) maintaining a queue of nodes in BFS and CaveBFS. These observations imply that the weak models' failure to perform well initially is one of the major reasons behind the performance gap between weak models and strong models. Therefore, even a limited series of correct steps at the beginning can significantly assist models in sequential reasoning tasks. Furthermore, for models possessing a sufficient level of sequential reasoning ability, this process may lead to a form of self-guidance, where the model reinforces its decisions based on prior correct actions.

## 5    Related Works

**Large Language Models** As the number of parameters and the scale of pretrained data increase, emerging behaviors become evident, allowing the model to perform tasks that are unachievable when complexity remains below a specified threshold (OpenAI, 2023). Additionally, it has been found that the utilization of meticulously crafted prompts (Fu et al., 2022; Zhou et al., 2022) can significantly improve the reasoning capabilities of LLMs. Open-source models (Touvron et al., 2023; Taori et al., 2023) have also emerged from community efforts and shown their effectiveness, with Vicuna (Chiang et al., 2023) and Mixtral (Jiang et al., 2024) demonstrating close-to-GPT-3.5 performance on human benchmarks. In our evaluation, we found that despite the impressive chat abilities, there exists a gap in the algorithmic reasoning abilities between open-source models and close-sourced models.

**Benchmarking Reasoning Abilities.** Benchmarking LLMs' reasoning abilities in performing complicated tasks have always been difficult. Most existing benchmarks in this area mainly focus on well-formulated objectives such as coding benchmarks like Codex (Chen et al., 2021), Swe-bench (Jimenez et al., 2023) and LiveCodeBench(Jain et al., 2024) and mathematical benchmarks like GSM8K (Cobbe et al., 2021), MMLU (Hendrycks et al., 2020),MATH (Hendrycks et al., 2021),MMLU(Wang et al., 2024). These benchmarks is based on one-round Q&A and lack the ability to access LLMs' sequential reasoning and planning capabilities. To address this, agentic benchmarks, *e.g.,* website agent benchmarks such as WebArena (Zhou et al., 2023), Mind2Web (Deng et al., 2023), VisualWebArena (Koh et al., 2024), *etc.,* have been devised to assess LLMs within real-life scenarios that inherently necessitate foresight and retrospective analysis. However, contrary to coding and mathematical problems, it is challenging for agentic benchmark to scale in size and to quantify the complexity of the environment, thereby impeding the systematic investigation of how these factors impact the performance of models. Furthermore, given the absence of known optimal strategies for these tasks, it proves difficult to guide LLMs and to examine how different prompting techniques (Wei et al., 2022; Zhou et al.) may affect the models' performance. Some works (Shi et al., 2023) investigate how these prompting techniques affect LLMs' reasoning ability but they didn't take planning and sequential reasoning ability into account and the resulted conclusions are sill superficial. Unlike previous works, our proposed AQA-Bench provides more comprehensive empirical evidence and results in deeper analysis.

## 6    Conclusion

In this study, we embark on an initial exploration into the realm of evaluating LLMs within interactive environments. These environments necessitate a deep understanding of specific algorithmic procedures by the LLMs, ranging from efficiently guessing a number within minimal steps to strategically searching for unvisited nodes in a graph. Our comprehensive evaluation reveals a notable performance gap between current open-source and closed-sourced models, with the latter showing superior capabilities in these tasks. We expect future efforts to focus on introducing a broader range of interactive environments and developing more effective prompting strategies to better equip LLMs for these benchmarks.

### Acknowledgment

This work is partially supported by Open Philanthropy. We thank the Microsoft Accelerate Foundation Models Research Program, the OpenAI Researcher Access Program, Google Cloud Research Credits Program, Center for AI Safety, and Lambda Cloud for supporting our computing needs.

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

# A    Effect of CoT prompting

Although Chain-of-Thought prompting have been proven to be highly effective in the reasoning-intensive tasks, such as coding and mathematical problems, we find that it can not consistently improve performance across all environments, as shown in Tabs. 5 and 6.

Table 5: **The comparison between evaluation results without and with Chain-of-Thought.** For models with strong goal metrics (*e.g.*, $Err_{min}$, $G_{min}$) indicting weak performance, goal metrics are more informative than policy metrics (*e.g.*, $Err_{sum}$, $G_{sum}$, ACC).

| | Base Envs under HARD Mode | | | | | | | | |
|---|---|---|---|---|---|---|---|---|---|
| Model | BinarySearch | | | DFS | | | BFS | | |
| | $Err_{min} \downarrow$ | $Err_{sum} \downarrow$ | ACC $\uparrow$ | $G_{min} \downarrow$ | $G_{sum} \downarrow$ | ACC $\uparrow$ | $G_{min} \downarrow$ | $G_{sum} \downarrow$ | ACC $\uparrow$ |
| Llama3-70B-Instruct | **0.00** | **1.20** | 0.00 | **0.25** | 8.12 | **0.56** | **0.02** | **11.10** | **0.37** |
| Llama3-70B-Instruct w/ CoT | **0.00** | 1.44 | **0.05** | 0.43 | **7.86** | 0.50 | 0.24 | 13.68 | 0.13 |

Table 6: **The comparison between evaluation results without and with Chain-of-Thought.** For models with strong goal metrics (*e.g.*, $Err_{min}$, $G_{min}$) indicting weak performance, goal metrics are more informative than policy metrics (*e.g.*, $Err_{sum}$, $G_{sum}$, ACC).

| | Embodied Envs under HARD Mode | | | | | | | | |
|---|---|---|---|---|---|---|---|---|---|
| Model | Coin | | | CaveDFS | | | CaveBFS | | |
| | $Err_{min} \downarrow$ | $Err_{sum} \downarrow$ | ACC $\uparrow$ | $G_{min} \downarrow$ | $G_{sum} \downarrow$ | ACC $\uparrow$ | $G_{min} \downarrow$ | $G_{sum} \downarrow$ | ACC $\uparrow$ |
| Llama3-70B-Instruct | **0.00** | 0.62 | 0.00 | **0.13** | 8.74 | **0.56** | **0.05** | **11.89** | **0.09** |
| Llama3-70B-Instruct w/ CoT | **0.00** | **0.61** | **0.03** | 0.28 | **7.53** | **0.56** | 0.14 | 13.51 | 0.08 |

### A.1 Re-productivity and Variance

Although test cases in our AQA-Bench can be generated dynamically, we pre-generated a test set with 400 test cases for each base environment under the EASY mode for simpler re-production. The final scores are averaged among test cases.

## B Variance of Test Sets

Given that GuessNum, DFS and BFS each can have at most 32768, $1.18 * 10^6$, $9.17 * 10^{16}$ test cases under the EASY mode, the quantity of our pre-generated test cases is somewhat modest. To verify that evaluation results with this number of test cases are valid and representative of the models' performance in each environment, we generated another 3 equally sized test sets and evaluated Llama2-7B-Chat and Vicuna-7B-v1.5-16K on all 4 test sets. To quantify the variance of results, we define that

$$\text{Avg} = \frac{\sum \{m_i\}}{|\{m_i\}|} \tag{6}$$

$$\text{Margin}_{\min} = \text{Avg} - \min(\{m_i\}) \tag{7}$$

$$\text{Margin}_{\max} = \max(\{m_i\}) - \text{Avg}, \tag{8}$$

where $\min(\{m_i\})$ is a set of the same metric from different evaluation runs. $\text{Margin}_{\min}$ and $\text{Margin}_{\max}$ can be viewed as a measurement for variance of evaluation results.

As shown in Tab. 7, $\text{Margin}_{\min}$ and $\text{Margin}_{\max}$ are relatively low compared to metric difference across models, which shows that evaluation results drawn from our pre-generated test set (with only 400 cases) can sufficiently represent the tested models' performance in environments with this level of complexity. Therefore, we only report results from the first test set rather than from all four test sets in the following context to save computation. For the HARD mode, we pre-generated 1500 test cases for each environment of which the variance is shown in in Tab. 8, indicating that under the HARD mode, the performance of the models do not show a strong variance on our test sets with 1500 test cases for each environments.

Table 7: **Inter-dataset variance of Llama2-7B-chat and Vicuna-7B-v1.5-16K under EASY mode.** Results are summarized from evaluations on 4 independently generated test sets. All tests are run with ICE=0 and no teacher-guiding.

**Base Environments**

| | GuessNum | | | DFS | | | BFS | | |
|---|---|---|---|---|---|---|---|---|---|
| | $\text{Err}_{\min}\downarrow$ | $\text{Err}_{\text{sum}}\downarrow$ | $\text{ACC}\uparrow$ | $\text{G}_{\min}\downarrow$ | $\text{G}_{\text{sum}}\downarrow$ | $\text{ACC}\uparrow$ | $\text{G}_{\min}\downarrow$ | $\text{G}_{\text{sum}}\downarrow$ | $\text{ACC}\uparrow$ |
| Llama2-7B-chat | | | | | | | | | |
| Avg | 0.265 | 7.895 | 0.000 | 0.598 | 3.588 | 0.235 | 0.605 | 9.531 | 0.002 |
| $\text{Margin}_{\min}$ | 0.009 | 0.185 | 0.000 | 0.017 | 0.115 | 0.016 | 0.008 | 0.274 | 0.001 |
| $\text{Margin}_{\max}$ | 0.006 | 0.168 | 0.000 | 0.020 | 0.138 | 0.010 | 0.007 | 0.269 | 0.001 |
| Vicuna-7B-v1.5-16K | | | | | | | | | |
| Avg | 0.476 | 9.606 | 0.000 | 0.644 | 5.769 | 0.151 | 0.849 | 10.067 | 0.029 |
| $\text{Margin}_{\min}$ | 0.017 | 0.366 | 0.000 | 0.009 | 0.196 | 0.007 | 0.010 | 0.640 | 0.001 |
| $\text{Margin}_{\max}$ | 0.016 | 0.341 | 0.000 | 0.006 | 0.181 | 0.012 | 0.006 | 0.324 | 0.001 |

**Embodied Environments**

| | Coin | | | CaveDFS | | | CaveBFS | | |
|---|---|---|---|---|---|---|---|---|---|
| | $\text{Err}_{\min}\downarrow$ | $\text{Err}_{\text{sum}}\downarrow$ | $\text{ACC}\uparrow$ | $\text{G}_{\min}\downarrow$ | $\text{G}_{\text{sum}}\downarrow$ | $\text{ACC}\uparrow$ | $\text{G}_{\min}\downarrow$ | $\text{G}_{\text{sum}}\downarrow$ | $\text{ACC}\uparrow$ |
| Llama2-7B-chat | | | | | | | | | |
| Avg | 0.079 | 5.256 | 0.000 | 0.488 | 4.633 | 0.340 | 0.757 | 5.405 | 0.046 |
| $\text{Margin}_{\min}$ | 0.005 | 0.238 | 0.000 | 0.008 | 0.060 | 0.011 | 0.001 | 0.206 | 0.002 |
| $\text{Margin}_{\max}$ | 0.008 | 0.269 | 0.000 | 0.011 | 0.021 | 0.010 | 0.001 | 0.255 | 0.001 |
| Vicuna-7B-v1.5-16K | | | | | | | | | |
| Avg | 1.000 | 1.000 | 0.000 | 0.538 | 7.684 | 0.208 | 0.717 | 13.914 | 0.069 |
| $\text{Margin}_{\min}$ | 0.000 | 0.000 | 0.000 | 0.015 | 0.389 | 0.007 | 0.007 | 0.403 | 0.004 |
| $\text{Margin}_{\max}$ | 0.000 | 0.000 | 0.000 | 0.013 | 0.361 | 0.006 | 0.009 | 0.473 | 0.003 |

### B.1 Variance of GPT models

Another factor that may affect our experimental conclusion is the randomness of the model itself. For open-source models and Gemini-Pro, we disable the random sampling in all the experiments. But for GPT models, as they can only be accessed via OpenAI API, we cannot turn off such model randomness. Here we evaluated GPT models on the same dataset for 4 times to investigate the variance of the GPT models' performance measurement.

It is shown in Tab. 9 that $\text{Margin}_{\min}$ and $\text{Margin}_{\max}$ are a lot smaller than metric difference between GPT-3.5-Turbo and GPT-4-Turbo. This demonstrates that 400 test cases are sufficient to alleviate the impact of GPTs' randomness on the evaluation results. Thus, the experimental results of GPTs we report in the following context are only from a single-time evaluation instead of summary of four different evaluations.

Table 8: **Inter-dataset variance of Llama2-7B-chat and Vicuna-7B-v1.5-16K under HARD mode.** Results are summarized from evaluations on 4 independently generated test sets. All tests are run with ICE=0 and no teacher-guiding.

**Base Environments**

| | GuessNum | | | DFS | | | BFS | | |
|---|---|---|---|---|---|---|---|---|---|
| | $\mathrm{Err_{min}}\downarrow$ | $\mathrm{Err_{sum}}\downarrow$ | $\mathrm{ACC}\uparrow$ | $\mathrm{G_{min}}\downarrow$ | $\mathrm{G_{sum}}\downarrow$ | $\mathrm{ACC}\uparrow$ | $\mathrm{G_{min}}\downarrow$ | $\mathrm{G_{sum}}\downarrow$ | $\mathrm{ACC}\uparrow$ |
| Llama2-7B-chat | | | | | | | | | |
| Avg | 0.498 | 14.956 | 0.000 | 0.735 | 7.360 | 0.193 | 0.761 | 15.949 | 0.005 |
| $\mathrm{Margin_{min}}$ | 0.006 | 0.187 | -0.000 | 0.004 | 0.319 | 0.003 | 0.003 | 0.670 | 0.002 |
| $\mathrm{Margin_{max}}$ | 0.011 | 0.332 | 0.000 | 0.007 | 0.397 | 0.007 | 0.006 | 0.390 | 0.002 |
| Vicuna-7B-v1.5-16K | | | | | | | | | |
| Avg | 0.476 | 9.606 | 0.000 | 0.644 | 5.769 | 0.151 | 0.849 | 10.067 | 0.029 |
| $\mathrm{Margin_{min}}$ | 0.008 | 0.012 | -0.000 | 0.002 | 0.205 | 0.002 | 0.002 | 0.302 | 0.000 |
| $\mathrm{Margin_{max}}$ | 0.006 | 0.022 | 0.000 | 0.006 | 0.119 | 0.001 | 0.003 | 0.479 | 0.001 |

**Embodied Environments**

| | Coin | | | CaveDFS | | | CaveBFS | | |
|---|---|---|---|---|---|---|---|---|---|
| | $\mathrm{Err_{min}}\downarrow$ | $\mathrm{Err_{sum}}\downarrow$ | $\mathrm{ACC}\uparrow$ | $\mathrm{G_{min}}\downarrow$ | $\mathrm{G_{sum}}\downarrow$ | $\mathrm{ACC}\uparrow$ | $\mathrm{G_{min}}\downarrow$ | $\mathrm{G_{sum}}\downarrow$ | $\mathrm{ACC}\uparrow$ |
| Llama2-7B-chat | | | | | | | | | |
| Avg | 0.079 | 5.256 | 0.000 | 0.488 | 4.633 | 0.340 | 0.757 | 5.405 | 0.046 |
| $\mathrm{Margin_{min}}$ | 0.005 | 0.238 | 0.000 | 0.008 | 0.060 | 0.011 | 0.001 | 0.206 | 0.002 |
| $\mathrm{Margin_{max}}$ | 0.008 | 0.269 | 0.000 | 0.011 | 0.021 | 0.010 | 0.001 | 0.255 | 0.001 |
| Vicuna-7B-v1.5-16K | | | | | | | | | |
| Avg | 1.000 | 1.000 | 0.000 | 0.538 | 7.684 | 0.208 | 0.717 | 13.914 | 0.069 |
| $\mathrm{Margin_{min}}$ | 0.000 | 0.000 | 0.000 | 0.015 | 0.389 | 0.007 | 0.007 | 0.403 | 0.004 |
| $\mathrm{Margin_{max}}$ | 0.000 | 0.000 | 0.000 | 0.013 | 0.361 | 0.006 | 0.009 | 0.473 | 0.003 |

Table 9: **Intra-dataset variance of GPT-3.5-Turbo and GPT-4-Turbo under EASY mode.** These results are summarized from results from 4 different test runs on the same test set. All tests are run with ICE=0 and no teacher-guiding.

**Base Environments**

| | GuessNum | | | DFS | | | BFS | | |
|---|---|---|---|---|---|---|---|---|---|
| | $\mathrm{Err_{min}}\downarrow$ | $\mathrm{Err_{sum}}\downarrow$ | $\mathrm{ACC}\uparrow$ | $\mathrm{G_{min}}\downarrow$ | $\mathrm{G_{sum}}\downarrow$ | $\mathrm{ACC}\uparrow$ | $\mathrm{G_{min}}\downarrow$ | $\mathrm{G_{sum}}\downarrow$ | $\mathrm{ACC}\uparrow$ |
| GPT-3.5-Turbo | | | | | | | | | |
| Avg | 0.000 | 0.513 | 0.003 | 0.348 | 5.142 | 0.618 | 0.116 | 6.773 | 0.517 |
| $\mathrm{Margin_{min}}$ | 0.000 | 0.003 | 0.002 | 0.006 | 0.078 | 0.006 | 0.002 | 0.096 | 0.004 |
| $\mathrm{Margin_{max}}$ | 0.000 | 0.004 | 0.003 | 0.003 | 0.064 | 0.006 | 0.002 | 0.141 | 0.008 |
| GPT-4-Turbo | | | | | | | | | |
| Avg | 0.000 | 0.496 | 0.493 | 0.025 | 3.935 | 0.935 | 0.002 | 6.087 | 0.383 |
| $\mathrm{Margin_{min}}$ | 0.000 | 0.000 | 0.033 | 0.004 | 0.025 | 0.003 | 0.000 | 0.004 | 0.016 |
| $\mathrm{Margin_{max}}$ | 0.000 | 0.000 | 0.012 | 0.003 | 0.016 | 0.004 | 0.000 | 0.005 | 0.018 |

**Embodied Environments**

| | Coin | | | CaveDFS | | | CaveBFS | | |
|---|---|---|---|---|---|---|---|---|---|
| | $\mathrm{Err_{min}}\downarrow$ | $\mathrm{Err_{sum}}\downarrow$ | $\mathrm{ACC}\uparrow$ | $\mathrm{G_{min}}\downarrow$ | $\mathrm{G_{sum}}\downarrow$ | $\mathrm{ACC}\uparrow$ | $\mathrm{G_{min}}\downarrow$ | $\mathrm{G_{sum}}\downarrow$ | $\mathrm{ACC}\uparrow$ |
| GPT-3.5-Turbo | | | | | | | | | |
| Avg | 0.001 | 1.013 | 0.000 | 0.199 | 4.769 | 0.669 | 0.272 | 9.595 | 0.100 |
| $\mathrm{Margin_{min}}$ | 0.001 | 0.017 | -0.000 | 0.001 | 0.085 | 0.010 | 0.007 | 0.106 | 0.002 |
| $\mathrm{Margin_{max}}$ | 0.001 | 0.008 | 0.000 | 0.002 | 0.103 | 0.010 | 0.005 | 0.059 | 0.003 |
| GPT-4-Turbo | | | | | | | | | |
| Avg | 0.000 | 0.496 | 0.506 | 0.237 | 3.503 | 0.755 | 0.118 | 8.071 | 0.161 |
| $\mathrm{Margin_{min}}$ | 0.000 | 0.000 | 0.007 | 0.007 | 0.069 | 0.011 | 0.003 | 0.052 | 0.002 |
| $\mathrm{Margin_{max}}$ | 0.000 | 0.000 | 0.007 | 0.008 | 0.117 | 0.008 | 0.007 | 0.059 | 0.002 |

# C   Full Evaluation Results with Policy Metrics

In this section, we present the full evaluation results with policy metrics.

## C.1   Main Results (ICE=0)

Table 10: **The main evaluation results with all environments.** For models with strong goal metrics (*e.g.*, $Err_{min}$, $G_{min}$) indicting weak performance, goal metrics are more informative than policy metrics (*e.g.*, $Err_{sum}$, $G_{sum}$, ACC).

**Base Envs under EASY Mode**

| Model | GuessNum | | | DFS | | | BFS | | |
|---|---|---|---|---|---|---|---|---|---|
| | $Err_{min}$ ↓ | $Err_{sum}$ ↓ | ACC ↑ | $G_{min}$ ↓ | $G_{sum}$ ↓ | ACC ↑ | $G_{min}$ ↓ | $G_{sum}$ ↓ | ACC ↑ |
| *Small < 10B* | | | | | | | | | |
| Llama2-7B-chat | 0.26 | 7.71 | 0.00 | 0.58 | 3.73 | 0.24 | 0.60 | 9.80 | 0.00 |
| Llama3-8B-Instruct | 0.01 | 1.14 | 0.00 | 0.21 | 4.66 | 0.51 | 0.02 | 6.68 | 0.23 |
| Vicuna-7B-v1.5-16K | 0.46 | 9.24 | 0.00 | 0.65 | 5.79 | 0.15 | 0.84 | 10.29 | 0.03 |
| Mistral-7B-Instruct-v02 | 0.06 | 2.02 | 0.00 | 0.49 | 2.72 | 0.61 | 0.24 | 8.72 | 0.13 |
| DeepSeek-LLM-7B | 0.43 | 9.24 | 0.00 | 0.34 | 6.59 | 0.36 | 0.52 | 11.20 | 0.06 |
| DeepSeek-MoE-16B | 1.00 | 1.00 | 0.00 | 0.63 | 4.78 | 0.07 | 0.88 | 8.18 | 0.02 |
| *10B ≤ Medium < 50B* | | | | | | | | | |
| Llama2-13B-chat | 0.01 | 3.24 | 0.00 | 0.34 | 5.98 | 0.41 | 0.65 | 10.59 | 0.05 |
| Vicuna-13B-v1.5-16K | 0.39 | 8.31 | 0.00 | 0.66 | 13.23 | 0.12 | 0.81 | 15.61 | 0.05 |
| Mixtral-8x7B-Instruct-v01 | **0.00** | 0.69 | 0.00 | 0.47 | **3.32** | 0.57 | 0.14 | 7.36 | 0.21 |
| DeepSeek-R1-Distill-Qwen-32B | 0.12 | 3.88 | 0.05 | 0.58 | 3.83 | 0.43 | 0.72 | 11.44 | 0.13 |
| *Large ≥ 50B* | | | | | | | | | |
| Llama2-70B-chat | 0.11 | 2.64 | 0.00 | 0.33 | 4.39 | 0.44 | 0.28 | 10.14 | 0.06 |
| Llama3-70B-Instruct | 0.01 | 1.41 | 0.00 | 0.10 | 4.37 | 0.68 | 0.01 | 6.08 | **0.53** |
| DeepSeek-LLM-67B | 0.12 | 5.62 | 0.00 | 0.40 | 4.34 | 0.42 | 0.45 | 11.59 | 0.09 |
| *Closed-source* | | | | | | | | | |
| GPT-3.5-Turbo | **0.00** | 0.51 | 0.01 | 0.35 | 5.21 | 0.61 | 0.11 | 6.68 | 0.52 |
| GPT-4-Turbo | **0.00** | **0.50** | **0.46** | 0.03 | 3.93 | 0.94 | 0.00 | 6.08 | 0.38 |
| Gemini-Pro | **0.00** | 0.63 | 0.00 | 0.25 | 3.71 | 0.76 | 0.06 | 7.39 | 0.17 |
| O1-Preview | **0.00** | **0.50** | 0.43 | **0.00** | 3.88 | **1.00** | **0.00** | **6.07** | **0.99** |

**Embodied Envs under EASY Mode**

| Model | Coin | | | CaveDFS | | | CaveBFS | | |
|---|---|---|---|---|---|---|---|---|---|
| | $Err_{min}$ ↓ | $Err_{sum}$ ↓ | ACC ↑ | $G_{min}$ ↓ | $G_{sum}$ ↓ | ACC ↑ | $G_{min}$ ↓ | $G_{sum}$ ↓ | ACC ↑ |
| *Small < 10B* | | | | | | | | | |
| Llama2-7B-chat | 0.07 | 5.02 | 0.00 | 0.50 | 4.65 | 0.33 | 0.76 | 5.66 | 0.05 |
| Llama3-8B-Instruct | 0.07 | 5.70 | 0.00 | 0.17 | 5.52 | 0.40 | 0.10 | 8.22 | 0.16 |
| Vicuna-7B-v1.5-16K | 1.00 | 1.00 | 0.00 | 0.54 | 8.04 | 0.21 | 0.72 | 14.39 | 0.07 |
| Mistral-7B-Instruct-v02 | 0.07 | 3.59 | 0.00 | 0.49 | 4.87 | 0.48 | 0.27 | 9.86 | 0.11 |
| DeepSeek-LLM-7B | 0.39 | 8.82 | 0.00 | 0.58 | 9.08 | 0.16 | 0.77 | 10.67 | 0.04 |
| DeepSeek-MoE-16B | 1.00 | 1.00 | 0.00 | 0.71 | **2.99** | 0.11 | 0.89 | **2.81** | 0.01 |
| *10B ≤ Medium < 50B* | | | | | | | | | |
| Llama2-13B-chat | 0.19 | 7.93 | 0.00 | 0.38 | 7.48 | 0.36 | 0.55 | 12.72 | 0.09 |
| Vicuna-13B-v1.5-16K | 1.00 | 1.00 | 0.00 | 0.56 | 8.18 | 0.21 | 0.64 | 11.28 | 0.06 |
| Mixtral-8x7B-Instruct-v01 | **0.00** | 0.78 | 0.00 | 0.32 | 4.61 | 0.45 | 0.15 | 8.48 | **0.17** |
| DeepSeek-R1-Distill-Qwen-32B | 0.18 | 5.99 | 0.00 | 0.72 | 11.44 | 0.13 | 0.58 | 3.83 | 0.44 |
| *Large ≥ 50B* | | | | | | | | | |
| Llama2-70B-chat | **0.00** | 0.51 | 0.00 | 0.35 | 4.53 | 0.44 | 0.30 | 10.51 | 0.03 |
| Llama3-70B-Instruct | 0.03 | 2.00 | 0.00 | 0.43 | 4.33 | **0.76** | 0.01 | 6.31 | **0.17** |
| DeepSeek-LLM-67B | 0.36 | 7.83 | 0.00 | 0.28 | 5.24 | 0.57 | 0.38 | 10.89 | 0.08 |
| *Closed-source* | | | | | | | | | |
| GPT-3.5-Turbo | **0.00** | 1.00 | 0.00 | 0.20 | 4.87 | 0.66 | 0.27 | 9.49 | 0.10 |
| GPT-4-Turbo | **0.00** | **0.50** | **0.50** | 0.33 | 3.62 | 0.74 | 0.12 | 8.07 | 0.16 |
| Gemini-Pro | **0.00** | 0.60 | 0.00 | 0.22 | 5.11 | 0.70 | 0.10 | 7.97 | 0.16 |
| O1-Preview | **0.00** | 0.79 | 0.05 | **0.00** | 3.88 | **1.00** | 0.12 | 8.88 | 0.08 |

**Base Envs under HARD Mode**

| Model | GuessNum | | | DFS | | | BFS | | |
|---|---|---|---|---|---|---|---|---|---|
| | $Err_{min}$ ↓ | $Err_{sum}$ ↓ | ACC ↑ | $G_{min}$ ↓ | $G_{sum}$ ↓ | ACC ↑ | $G_{min}$ ↓ | $G_{sum}$ ↓ | ACC ↑ |
| *Small < 10B* | | | | | | | | | |
| Llama2-7B-chat | 0.49 | 14.77 | 0.00 | 0.74 | 7.24 | 0.19 | 0.76 | 16.34 | 0.01 |
| Llama3-8B-Instruct | 0.07 | 6.64 | 0.00 | 0.41 | 7.90 | 0.43 | 0.07 | 12.59 | 0.13 |
| Vicuna-7B-v1.5-16K | 0.24 | 14.98 | 0.00 | 0.78 | 10.97 | 0.10 | 0.89 | 17.16 | 0.02 |
| Mistral-7B-Instruct-v02 | 0.06 | 3.43 | 0.00 | 0.65 | 4.11 | 0.61 | 0.46 | 16.29 | 0.08 |
| DeepSeek-LLM-7B | 0.49 | 6.42 | 0.00 | 0.61 | 16.07 | 0.18 | 0.71 | 19.62 | 0.04 |
| DeepSeek-MoE-16B | 1.00 | 1.00 | 0.00 | 0.78 | 8.96 | 0.03 | 0.92 | 11.38 | 0.01 |
| *10B ≤ Medium < 50B* | | | | | | | | | |
| Llama2-13B-chat | 0.49 | 14.77 | 0.00 | 0.59 | 11.21 | 0.25 | 0.76 | 17.27 | 0.03 |
| Vicuna-13B-v1.5-16K | 0.49 | 14.77 | 0.00 | 0.80 | 20.45 | 0.07 | 0.83 | 24.92 | 0.03 |
| Mixtral-8x7B-Instruct-v01 | **0.00** | 1.46 | 0.00 | 0.64 | **4.69** | 0.58 | 0.32 | 13.49 | 0.13 |
| DeepSeek-R1-Distill-Qwen-32B | 0.19 | 7.25 | 0.07 | 0.72 | 5.72 | 0.43 | 0.83 | 13.60 | 0.13 |
| *Large ≥ 50B* | | | | | | | | | |
| Llama2-70B-chat | 0.49 | 14.77 | 0.00 | 0.48 | 9.01 | 0.35 | 0.43 | 18.65 | 0.04 |
| Llama3-70B-Instruct | **0.00** | 1.20 | 0.00 | 0.25 | 8.12 | 0.56 | 0.02 | 11.10 | 0.37 |
| DeepSeek-LLM-67B | **0.00** | 0.70 | 0.00 | 0.51 | 9.48 | 0.28 | 0.67 | 21.98 | 0.05 |
| *Closed-source* | | | | | | | | | |
| GPT-3.5-Turbo | **0.00** | 0.59 | 0.00 | 0.55 | 8.76 | 0.51 | 0.27 | 13.30 | 0.29 |
| GPT-4-Turbo | **0.00** | **0.52** | 0.04 | 0.08 | 7.71 | 0.87 | 0.01 | 11.14 | 0.26 |
| Gemini-Pro | **0.00** | 0.81 | 0.00 | 0.33 | 7.36 | 0.69 | 0.12 | 13.59 | 0.09 |
| O1-Preview | **0.00** | 0.55 | **0.22** | **0.00** | 7.88 | **0.99** | **0.00** | 11.05 | **0.96** |

**Embodied Envs under HARD Mode**

| Model | Coin | | | CaveDFS | | | CaveBFS | | |
|---|---|---|---|---|---|---|---|---|---|
| | $Err_{min}$ ↓ | $Err_{sum}$ ↓ | ACC ↑ | $G_{min}$ ↓ | $G_{sum}$ ↓ | ACC ↑ | $G_{min}$ ↓ | $G_{sum}$ ↓ | ACC ↑ |
| *Small < 10B* | | | | | | | | | |
| Llama2-7B-chat | 0.49 | 14.77 | 0.00 | 0.68 | 9.78 | 0.19 | 0.83 | 12.04 | 0.04 |
| Llama3-8B-Instruct | 0.07 | 10.01 | 0.00 | 0.29 | 10.92 | 0.29 | 0.22 | 15.22 | 0.09 |
| Vicuna-7B-v1.5-16K | 0.49 | 14.77 | 0.00 | 0.70 | 15.89 | 0.13 | 0.83 | 24.45 | 0.05 |
| Mistral-7B-Instruct-v02 | 0.08 | 5.30 | 0.00 | 0.61 | 6.96 | 0.50 | 0.49 | 18.45 | 0.07 |
| DeepSeek-LLM-7B | 0.49 | 1.98 | 0.00 | 0.74 | 16.02 | 0.11 | 0.86 | 17.20 | 0.03 |
| DeepSeek-MOE-16B | 1.00 | 1.00 | 0.00 | 0.86 | **2.74** | 0.05 | 0.94 | **2.65** | 0.01 |
| *10B ≤ Medium < 50B* | | | | | | | | | |
| Llama2-13B-chat | 0.08 | 10.73 | 0.00 | 0.56 | 13.56 | 0.28 | 0.68 | 22.20 | 0.06 |
| Vicuna-13B-v1.5-16K | 1.00 | 1.00 | 0.00 | 0.65 | 14.78 | 0.17 | 0.71 | 20.33 | 0.05 |
| Mixtral-8x7B-Instruct-v01 | 0.07 | 2.22 | 0.00 | 0.50 | 8.21 | 0.38 | 0.30 | 15.46 | 0.09 |
| DeepSeek-R1-Distill-Qwen-32B | 0.19 | 7.25 | 0.07 | 0.73 | 6.73 | 0.38 | 0.84 | 14.21 | 0.13 |
| *Large ≥ 50B* | | | | | | | | | |
| Llama2-70B-chat | 0.08 | 13.72 | 0.00 | 0.49 | 9.35 | 0.33 | 0.46 | 18.94 | 0.02 |
| Llama3-70B-Instruct | **0.00** | 0.62 | 0.00 | 0.13 | 8.74 | 0.56 | **0.05** | **11.89** | **0.09** |
| DeepSeek-LLM-67B | 0.02 | 2.15 | 0.00 | 0.39 | 10.75 | 0.40 | 0.56 | 20.05 | 0.06 |
| *Closed-source* | | | | | | | | | |
| GPT-3.5-Turbo | 0.37 | 4.81 | 0.00 | 0.33 | 9.98 | 0.56 | 0.45 | 17.51 | 0.07 |
| GPT-4-Turbo | **0.00** | **0.52** | 0.04 | 0.33 | 7.04 | 0.67 | 0.19 | 14.67 | **0.09** |
| Gemini-Pro | **0.00** | 1.08 | 0.00 | 0.35 | 10.00 | 0.56 | 0.23 | 15.28 | **0.09** |
| O1-Preview | **0.00** | 0.67 | **0.22** | **0.00** | 7.88 | **0.99** | 0.22 | 16.16 | 0.05 |

## C.2   Effect of In-Context Examples (ICE=7)

Table 11: The evaluation results (ICE=7) in 3 base environments. For models with strong goal metrics (*e.g.*, $Err_{min}$, $G_{min}$) indicting weak performance, goal metrics are more informative than policy metrics (*e.g.*, $Err_{sum}$, $G_{sum}$, ACC).

| Model | GuessNum | | | DFS | | | BFS | | |
|---|---|---|---|---|---|---|---|---|---|
| | $Err_{min} \downarrow$ | $Err_{sum} \downarrow$ | ACC $\uparrow$ | $G_{min} \downarrow$ | $G_{sum} \downarrow$ | ACC $\uparrow$ | $G_{min} \downarrow$ | $G_{sum} \downarrow$ | ACC $\uparrow$ |
| Small < 10B | | | | | | | | | |
| Llama2-7B-chat | 0.08 (-0.18) | 2.32 (-5.39) | 0.10 (+0.10) | 0.39 (-0.19) | 9.04 (+5.31) | 0.23 (-0.01) | 0.65 (+0.05) | 8.28 (-1.52) | 0.14 (+0.14) |
| Llama3-8B-Instruct | 0.02 (+0.01) | 1.31 (+0.17) | 0.19 (+0.19) | 0.04 (-0.17) | 5.38 (+0.72) | 0.48 (-0.03) | 0.37 (+0.35) | 9.89 (+3.21) | 0.12 (-0.11) |
| Vicuna-7B-v1.5-16K | 0.02 (-0.44) | 1.27 (-7.97) | 0.22 (+0.22) | 0.37 (-0.28) | 8.61 (+2.82) | 0.27 (+0.12) | 0.68 (-0.16) | 13.10 (+2.81) | 0.15 (+0.12) |
| Mistral-7B-Instruct-v02 | 0.01 (-0.05) | 1.07 (-0.95) | 0.22 (+0.22) | 0.14 (-0.35) | 5.74 (+3.02) | 0.51 (-0.10) | 0.39 (+0.15) | 9.87 (+1.15) | 0.17 (+0.04) |
| DeepSeek-LLM-7B | 0.04 (-0.39) | 1.50 (-7.74) | 0.18 (+0.18) | 0.16 (-0.18) | 6.93 (+0.34) | 0.17 (-0.19) | 0.61 (+0.09) | 11.43 (+0.23) | 0.18 (+0.12) |
| DeepSeek-MoE-16B | 0.02 (-0.98) | 1.51 (+0.51) | 0.21 (+0.21) | 0.14 (-0.49) | 6.75 (+1.97) | 0.30 (+0.23) | 0.86 (-0.02) | **2.60** (-5.58) | 0.10 (+0.08) |
| 10B ≤ Medium < 50B | | | | | | | | | |
| Llama2-13B-chat | 0.06 (+0.05) | 1.89 (-1.35) | 0.13 (+0.13) | 0.50 (+0.16) | 10.75 (+4.77) | 0.18 (-0.23) | 0.57 (-0.08) | 11.48 (+0.89) | 0.09 (+0.04) |
| Vicuna-13B-v1.5-16K | 0.12 (-0.27) | 5.42 (-2.89) | 0.12 (+0.12) | 0.16 (-0.50) | 5.24 (-7.99) | 0.63 (+0.51) | 0.23 (-0.58) | 8.14 (-7.47) | 0.27 (+0.22) |
| Mixtral-8x7B-Instruct-v01 | **0.00** (+0.00) | 0.56 (-0.13) | 0.25 (+0.25) | 0.20 (-0.27) | 6.46 (+3.14) | 0.44 (-0.13) | 0.48 (+0.34) | 11.34 (+3.98) | 0.21 (+0.00) |
| Large ≥ 50B | | | | | | | | | |
| Llama2-70B-chat | 0.07 (-0.04) | 1.96 (-0.68) | 0.13 (+0.13) | 0.14 (-0.19) | 5.88 (+1.49) | 0.46 (+0.02) | 0.46 (+0.18) | 9.46 (-0.68) | 0.11 (+0.05) |
| Llama3-70B-Instruct | 0.01 (+0.00) | 0.70 (-0.71) | 0.19 (+0.19) | **0.00** (-0.10) | 4.30 (-0.07) | 0.72 (+0.04) | **0.00** (-0.01) | 6.37 (+0.29) | 0.37 (-0.16) |
| DeepSeek-LLM-67B | **0.00** (-0.12) | 0.58 (-5.04) | 0.25 (+0.25) | 0.18 (-0.22) | 6.79 (+2.45) | 0.33 (-0.09) | 0.36 (-0.09) | 10.40 (-1.19) | 0.18 (+0.09) |
| Closed-source | | | | | | | | | |
| GPT-3.5-Turbo | **0.00** (-0.00) | 0.52 (+0.01) | 0.01 (+0.00) | 0.36 (+0.01) | 5.30 (+0.09) | 0.62 (+0.01) | 0.12 (+0.01) | 6.63 (-0.05) | **0.51** (-0.01) |
| GPT-4-Turbo | **0.00** (-0.00) | **0.50** (+0.00) | **0.47** (+0.01) | 0.02 (-0.01) | **3.93** (+0.00) | **0.94** (+0.00) | **0.00** (+0.00) | **6.08** (-0.00) | 0.40 (+0.02) |
| Gemini-Pro | **0.00** (+0.00) | 0.51 (-0.12) | 0.43 (+0.43) | 0.02 (-0.23) | 4.57 (+0.86) | 0.68 (-0.08) | 0.03 (-0.03) | 6.59 (-0.80) | 0.36 (+0.19) |

Table 12: The evaluation results (ICE=7) in 3 embodied environments. For models with strong goal metrics (*e.g.*, $Err_{min}$, $G_{min}$) indicting weak performance, goal metrics are more informative than policy metrics (*e.g.*, $Err_{sum}$, $G_{sum}$, ACC).

| Model | Coin | | | CaveDFS | | | CaveBFS | | |
|---|---|---|---|---|---|---|---|---|---|
| | $Err_{min} \downarrow$ | $Err_{sum} \downarrow$ | ACC $\uparrow$ | $G_{min} \downarrow$ | $G_{sum} \downarrow$ | ACC $\uparrow$ | $G_{min} \downarrow$ | $G_{sum} \downarrow$ | ACC $\uparrow$ |
| Small < 10B | | | | | | | | | |
| Llama2-7B-chat | 0.11 (+0.04) | 2.70 (-2.32) | 0.08 (+0.08) | 0.38 (-0.12) | 8.79 (+4.14) | 0.26 (-0.07) | 0.58 (-0.18) | 11.27 (+5.61) | 0.11 (+0.06) |
| Llama3-8B-Instruct | 0.02 (-0.05) | 1.22 (-4.48) | 0.19 (+0.19) | 0.04 (-0.13) | 5.37 (-0.15) | 0.49 (+0.09) | 0.42 (+0.32) | 10.12 (+1.90) | 0.10 (-0.06) |
| Vicuna-7B-v1.5-16K | 0.02 (-0.98) | 1.13 (+0.13) | 0.22 (+0.22) | 0.39 (-0.15) | 8.88 (+0.84) | 0.25 (+0.04) | 0.68 (-0.04) | 13.48 (-0.91) | 0.14 (+0.07) |
| Mistral-7B-Instruct-v02 | 0.01 (-0.06) | 1.15 (-2.44) | 0.22 (+0.22) | 0.18 (-0.31) | 6.28 (+1.41) | 0.45 (-0.03) | 0.48 (+0.21) | 10.24 (+0.38) | 0.15 (+0.04) |
| DeepSeek-llm-7B | 0.04 (-0.35) | 1.53 (-7.29) | 0.17 (+0.17) | 0.19 (-0.39) | 6.80 (-2.28) | 0.33 (+0.17) | 0.62 (-0.15) | 12.06 (+1.39) | 0.17 (+0.13) |
| DeepSeek-moe-16B | 0.02 (-0.98) | 1.61 (+0.61) | 0.21 (+0.21) | 0.13 (-0.58) | 6.71 (+3.72) | 0.34 (+0.23) | 0.87 (-0.02) | **2.71** (-0.10) | 0.08 (+0.07) |
| 10B ≤ Medium < 50B | | | | | | | | | |
| Llama2-13B-chat | 0.05 (-0.14) | 1.98 (-5.95) | 0.13 (+0.13) | 0.48 (+0.10) | 10.50 (+3.02) | 0.19 (-0.17) | 0.56 (+0.01) | 11.65 (-1.07) | 0.11 (+0.02) |
| Vicuna-13B-v1.5-16K | 0.13 (-0.87) | 5.48 (+4.48) | 0.12 (+0.12) | 0.15 (-0.41) | 5.50 (-2.68) | 0.59 (+0.38) | 0.27 (-0.37) | 8.54 (-2.74) | 0.27 (+0.21) |
| Mixtral-8x7B-Instruct-v01 | **0.00** (-0.00) | 0.64 (-0.14) | 0.23 (+0.23) | 0.17 (-0.15) | 6.27 (+1.66) | 0.43 (-0.02) | 0.39 (+0.24) | 10.40 (+1.92) | 0.21 (+0.04) |
| Large ≥ 50B | | | | | | | | | |
| Llama2-70B-chat | 0.09 (+0.09) | 2.37 (+1.86) | 0.12 (+0.12) | 0.20 (-0.15) | 6.66 (+2.13) | 0.42 (-0.02) | 0.60 (+0.30) | 8.39 (-2.12) | 0.08 (+0.05) |
| Llama3-70B-Instruct | 0.01 (-0.02) | 0.73 (-1.27) | 0.17 (+0.17) | **0.00** (-0.03) | 4.23 (-0.10) | 0.75 (-0.01) | **0.01** (+0.00) | **6.41** (+0.10) | 0.32 (+0.15) |
| DeepSeek-llm-67B | **0.00** (-0.36) | 0.57 (-7.26) | 0.24 (+0.24) | 0.18 (-0.10) | 6.67 (+1.43) | 0.37 (-0.20) | 0.39 (+0.01) | 10.54 (-0.35) | 0.21 (+0.13) |
| Closed-source | | | | | | | | | |
| GPT-3.5-Turbo | 0.02 (+0.02) | 1.02 (+0.02) | 0.00 (+0.00) | 0.19 (-0.01) | 4.83 (-0.04) | 0.66 (+0.00) | 0.27 (-0.00) | 9.56 (+0.07) | 0.10 (+0.00) |
| GPT-4-Turbo | **0.00** (-0.00) | **0.50** (-0.00) | **0.50** (+0.00) | 0.23 (-0.00) | **3.49** (-0.13) | **0.76** (+0.02) | 0.11 (-0.01) | 8.07 (-0.00) | 0.16 (+0.00) |
| Gemini-Pro | **0.00** (+0.00) | 0.51 (-0.09) | 0.41 (+0.41) | 0.04 (-0.18) | 5.17 (+0.06) | 0.54 (-0.16) | 0.05 (-0.05) | 6.79 (-1.18) | **0.33** (+0.17) |

# D    Prompt Instructions for the Models

In this section, we present the prompts we fed to the models.

## D.1    Base Environments

> **GuessNum**
>
> You are required to guess the random number which I have just picked between {min} and {max}. I will only tell you whether the true number is bigger or lower than your guess. Adjust your guess according to my response. Try as few times as you can. You can only reply with an integer number between {min} and {max}.

> **DFS**
>
> You are required to visit all the nodes in an undirected non-cyclic graph. An undirected non-cyclic graph contains a set of nodes and a set of edges that each connect a pair of nodes. All edges are undirected so that you can move from one node to the other connected by the edge in either direction. Every time you visit a node, you will be given the adjacent nodes connected to this node. You can only reply with an integer number indicating which node to be visited next. Do not explain your answer. Try to traverse the entire graph in as few rounds as possible. You are currently on the node 0. You should use depth-first-search algorithm, each time you should select a node you have not moved to. If all nodes adjacent to the current node have been visited, you should backtrack to the node through which you entered this node for the first time.

> **BFS**
>
> You are required to visit all the nodes in an undirected non-cyclic graph. An undirected non-cyclic graph contains a set of nodes, and a set of edges that each connects a pair of nodes. Every time you visit a node, you will be given the adjacent nodes connected to this node. You can only visit nodes that are adjacent to the already visited nodes. You can only reply with an integer number indicating which node to be visited next. Do not explain your answer. Try to traverse the entire graph in as few rounds as possible. You are currently on the node 0. You should use breadth-first-search algorithm. The algorithm works as follows: 1. Initialize a queue data structure and add the starting node to the queue. 2. While the queue is not empty, visit the first node and remove it from the queue. 3. For nodes adjacent to the removed vertex, add the unvisited ones to the queue. 4. Repeat steps 2-3 until the queue is empty.

## D.2    Embodied Environments

> **Coin**
>
> You are in a hidden temple where an old witch sits with a chest of gold. The witch promises to reward you with gold coins, the amount hidden within the chest ranging from {min} and {max}. To claim your prize, you must correctly guess the exact number of gold coins in the chest. After each guess, the witch will hint if the actual amount is higher or lower than your guess. Use these clues to adjust your guess accordingly. Try as few times as you can. You can only reply with an integer number between {min} and {max}.

> **CaveDFS**
>
> There is an expansive underground cave system in which each cave is uniquely numbered and interconnected by tunnels. Every time you visit a cave, you will know the adjacent caves directly connected to this one. You can only reply with an integer number indicating which cave to be visited next. Do not explain your answer. Your objective is to explore every cave, starting from cave 0. Try to visit all the caves in as few rounds as possible. You are currently in the cave 0.

> **CaveBFS**
>
> There is an expansive underground cave system in which each cave is uniquely numbered and interconnected by tunnels. Every time you and your team visit a cave, you will know the adjacent caves directly connected tno this one. Your team will then split into smaller groups to explore different caves, but groups can only move to caves adjacent to the visited cave. You can only reply with an integer number indicating which cave to be visited next. Do not explain your answer. Your objective is to explore every cave, starting from cave 0. Try to visit all the caves in as few rounds as possible. You and your team are currently in the cave 0.

## E  Model Versions

We used the checkpoint '2023-11-06' for GPT-3.5 and GPT-4.0 and '2024-09-12' for O1-Preview. The open-source model and the corresponding commit ID on HuggingFace are listed as below

- **Llama2-7B-chat** c1b0db933684edbfe29a06fa47eb19cc48025e93

- **Llama2-13B-chat** c2f3ec81aac798ae26dcc57799a994dfbf521496

- **Llama2-70B-chat** e1ce257bd76895e0864f3b4d6c7ed3c4cdec93e2

- **Llama3-8B-Instruct** e1945c40cd546c78e41f1151f4db032b271faeaa

- **Llama3-70B-Instruct** c6beed86a45dc2c96b11c2b2daaffd2c40d80cec

- **Vicuna-7B-v1.5-16K** c8df3ca4436a3bce5c4b5877e0117032081852b4

- **Vicuna-13B-v1.5-16K** 17c61f9ca19f5a7a04e96b2cc0d9bcf2920cb8c2

- **Mistral-7B-Instruct-v0.2** b70aa86578567ba3301b21c8a27bea4e8f6d6d61

- **Mixtral-8x7B-Instruct-v0.1** 125c431e2ff41a156b9f9076f744d2f35dd6e67a

- **DeepSeek-LLM-7B** afbda8b347ec881666061fa67447046fc5164ec8

- **DeepSeek-LLM-67B** 79648bef7658bb824e4630740f6e1484c1b0620b

- **DeepSeek-MoE-16B** cc01c87767bd905af4cb364693fd107014694ab9

- **DeepSeek-R1-Distill-Qwen-32B** 711ad2ea6aa40cfca18895e8aca02ab92df1a746

