# OpenReview forum: "AQA-Bench: An Interactive Benchmark for Evaluating LLMs’ Sequential Reasoning Ability in Algorithmic Environments"
_TMLR — Accepted by TMLR_

### Review · Reviewer_yQcR · 2025-02-01

**Summary Of Contributions:**

This work (i) provides a benchmark AQA-bench which enables the evaluation of LLMs’ sequential reasoning capabilities in interactive environments, (ii) it evaluates a wide set of LLMs and prompting strategies on this benchmark.

**Audience:**

Yes

**Broader Impact Concerns:**

I have no concerns.

**Claims And Evidence:**

Yes

**Requested Changes:**

1. Provide more information about arguments specified in Weakness 1.

2. Revise prompts to make them clearer.

**Strengths And Weaknesses:**

Strengths:

1. This paper provides a new data source for interactive sequential reasoning evaluation.

2. The paper is well-structured and easy to follow.
3. This work covers the evaluation of a wide range of closed- and open-source models, which provides comprehensive insights for future works.
4. It’s interesting to see that scaling does not always bring better performance.

Weaknesses:

1. **Some arguments are a bit unclear or over-claimed**: On page 1, for instance, the authors claim that existing works ‘do not assess the models’ capacity for procedural adherence, active memory maintenance, and ability to think ahead, which are elements vital for more complex, sequential reasoning tasks’, while this is not entirely accurate. There are already existing works on code simulation, which evaluate LLMs’ algorithmic capability (e.g., [1]), and works that use LLMs as embodied agents to interact with the environment (e.g., [2]). These tasks require the abilities highlighted by the authors as well. Moreover, some insights provided in the paper are also widely observed in previous papers: e.g., ICL is well known to be brittle to the design of the in-context examples (e.g., [3]). Can the authors make more concrete claims about the novel contributions of this work?

2. **Insufficient prompt design**: The prompt, in my opinion, is not well-designed: it is not even clear enough for humans to understand what to do. For instance, in the guess number task, the starting instruction is mere “You are required to guess a number between 0 and 8.” To specify the task, more information about the task objective and how the environment provides feedback should also be specified (e.g., ‘At each round, you will only give me a number of your guess. I will give you feedback about whether your guess is smaller/bigger than or equal to the target number. You should use as few rounds of guessing as possible.’)

[1] Liu, Changshu, Shizhuo Dylan Zhang, and Reyhaneh Jabbarvand. "Codemind: A framework to challenge large language models for code reasoning." arXiv preprint arXiv:2402.09664 (2024).

[2] Song, Chan Hee, et al. "Llm-planner: Few-shot grounded planning for embodied agents with large language models." Proceedings of the IEEE/CVF International Conference on Computer Vision. 2023.

[3] Shi, Freda, et al. "Large language models can be easily distracted by irrelevant context." International Conference on Machine Learning. PMLR, 2023.

---

> ### Author Response · Authors · 2025-03-22
>
> We thank the reviewer for the detailed comments and agree that our proposed benchmark covers a wide range of models and our findings are insightful and comprehensive. We address your concerns as follows:
>
> ## Comparision with previous works
> Although Shi et al. [3]. have also investigated the effect of in-context examples on LLMs, our benchmark provided more comprehensive evidence with different tested models and different numbers of in-context examples, such as the relationship between the negative effect of in-context examples and model sizes. This leads to a more in-depth analysis that the negative effect of in-context exemplars results from the overfitting of LLMs' in-context learning process, as larger models with stronger ICL capabilities are more pruned to examples' negative effect, while smaller models can still benefit from in-context exemplars as the number of exemplars increases, which contradicts to the conclusion from Shi et al.
>
> As for the comparison with Liu et al. [1]. and Song et al.[2], the purpose of our benchmark is not to assess LLMs' capability in solving algorithmic problems and interacting with environments but rather to use algorithmic environments to systematically study LLMs' sequential reasoning ability and how this ability is affected by various factors and external guidance. The motivation behind the usage of algorithmic environments, as stated in Sec. 2.1, is for 1) scalable dataset size, 2) controllable environment complexity, and 3) known optimal policy for more straightforward analysis and providing guidance.
>
> We further clarified this in Sec. 5 of our revised paper.
>
> ## Prompts are hard to understand
> "You are required to guess a number between xxx and yyy" as the prompt for GuessNum is only for a more straightforward demonstration in Fig. 1 and 2. The full version of our prompt is listed in Appendix D.
>
> Also, as shown in Tab. 3, proprietary models such as GPT-4 and Gemini-Pro are quite capable of solving the given tasks within the step number limitation, as they show extremely low goal metrics across environments. Therefore, we determine that the prompts we designed are already clear enough to access models' performance under various settings.
>
> [1] Liu, Changshu, Shizhuo Dylan Zhang, and Reyhaneh Jabbarvand. "Codemind: A framework to challenge large language models for code reasoning." arXiv preprint arXiv:2402.09664 (2024).
>
> [2] Song, Chan Hee, et al. "Llm-planner: Few-shot grounded planning for embodied agents with large language models." Proceedings of the IEEE/CVF International Conference on Computer Vision. 2023.
>
> [3] Shi, Freda, et al. "Large language models can be easily distracted by irrelevant context." International Conference on Machine Learning. PMLR, 2023.

---

### Review · Reviewer_TTaH · 2025-02-17

**Summary Of Contributions:**

The key contribution of this paper is AQA-Bench, a new benchmark to evaluate LLMs' sequential reasoning abilities in algorithmic games. The benchmark is interactive: e.g., in DFS games, the LLM will see adjacent nodes once it reaches a node, and it has to remember visited nodes and plans the next move accordingly. The paper evaluates 14 LLMs, and show interesting findings: 1) close-source models are better than open-source models; 2) in-context examples hurt performance; 3) predecessor steps (following optimal policy) as few-shot examples work better than steps of other cases (following optimal policy); 4) weak models struggle to start well; 5) scaling up doesn't seem to help much in this setting.

**Audience:**

Yes

**Broader Impact Concerns:**

I have no concerns on the ethical implications of the work.

**Claims And Evidence:**

Yes

**Requested Changes:**

Could you run experiments with CoT reasoning to generate actions at each step?

Particularly, it will be very interesting to see how CoT reasoning affects your current conclusions (2) - (5).

Can you also share your thoughts on including other algorithms into AQA-Bench? Something more contextual and may have a learnable heuristics to guide search and planning?

**Strengths And Weaknesses:**

**Strengths**

The benchmark is well scoped.

There are extensive experiments on a range of LLMs.

Experimental findings are interesting.

Paper writing is clear.

**Weaknesses**

I am not quite sure about the impact of this work.
- Algorithmic games are interesting, but they are not representatives of real-life sequential decision making problems that we encounter in reality.
- Algorithmic games can be hard, but it does not seem to be a good testbed for "sequential reasoning". Specially, for binary search, optimal actions are always to choose the new middle point; for DFS, it is optimal to always go deeper and then backtrack; for BFS, it is optimal to always traverse the graph layer by layer. It doesn't seem that one really needs to *plan* at all in these games: we don't need to think about what future information an action may reveal; we aren't allowed to undo or redo a previous action either. (Note: here my "undo" and "redo" are different from the backtracking in DFS.)

I like the findings of the paper, but I am concerned that some findings are not as deep as it should reach.
- When DFS/BFS algorithm description is given, the problem seems to become "which LLMs can follow algorithmic instructions well" (again, not "sequential reasoning" on-the-fly as the benchmark aims to test).
- Since picking an optimal action requires some reasoning about visited nodes and trajectories, it seems natural to use chain-of-thought (CoT) reasoning methods to generate the action at each step, but the paper hasn't done experiments of this kind.

---

> ### Author Response · Authors · 2025-03-22
>
> We thank the reviewer for the detailed comments and find that our benchmark is well scoped with extensive emprical evidence and interesting findings. We address your concerns as follows:
>
> ## Motivation behind algorithmic environments
> First of all, the algorithmic environments we proposed are based on common and useful algorithms such as Binary Search and traverse algorithms on graphs, which are easily encountered in real-life scenarios, even more easily than the scenarios that many common benchmarks access LLMs' knowledge in expert areas such as medicine.
>
> Also, as the algorithmic problems don't directly appear in an abstract way, such as traversing a graph in the fewest steps, we included 3 embodied environments to disguise the need for performing an algorithm with more real-life scenarios.
>
> Furthermore, as stated in Sec. 2.1, our main motivation behind the usage of algorithmic environments is for 1) scalable dataset size, 2) controllable environment complexity, and 3) known optimal policy for more straightforward analysis and providing guidance.
>
> Finally, one may find that planning is not really required in these environments because the optimal action at each current step is known for a human, which is by design. However, these algorithms are essentially highly formulated ways to plan a solution for each certain objective, and these algorithms are optimal precisely because they take future information into
>
> ## Prompts for DFS/BFS environments
> Although detailed instructions for DFS and BFS environments are given, these instructions are not provided in the prompts for their embodied variants, e.g., CaveDFS and CaveBFS. The conclusions drawn from these embodied environments are consistent with those from base DFS and BFS environments.
>
> ## Additional experiments
> For experiments with CoT,  we provide the evaluation results of Llama3-70B with CoT on embodied environments in Appendix A in our revised version, showing that CoT is not consistently helpful across different environments.
>
> As for experiments with extra algorithms, we find that the current settings are already sufficient to make insightful observations, so we have no current intention of including more algorithms in this version.

---

> > ### Author Response · Authors · 2025-03-24
> >
> > ## Updated CoT Experiments with Llama3-70B
> > We have updated the evaluation results of Llama3-70B with CoT on base environments in addition to embodied environments in Appendix A. The conclusion drawn from these results is consistent with the previous one that CoT is not consistently helpful across different environments.

---

### Review · Reviewer_ZLgk · 2025-02-25

**Summary Of Contributions:**

This paper presents a new benchmark, AQA-bench, which seeks to measure the ability of language models to reason about sequential tasks. The benchmark consists of three core algorithmic tasks: a "20 questions" style number guessing game with binary (higher/lower) feedback; a depth-first DAG exploration task; and a breadth-first variation of the same. The authors also construct a more naturalistic version of each task, which targets the same underlying mechanisms but in a "gamified" manner, to see whether this affects the models' ability to reason optimally. Using this benchmark, the authors evaluate a broad range of commercial and open models, finding that the commercial models tend to do significantly better. However, further experiments reveal that the differences are reduced with sufficient in-context examples. Finally, the authors argue that their results contradict common wisdom regarding how performance is expected to scale with model size, as they do not find that larger models always perform the best out of their model family.

**Audience:**

Yes

**Broader Impact Concerns:**

None.

**Claims And Evidence:**

Yes

**Requested Changes:**

- Consider making the relationship to agentic benchmarks such as webarena, SWE-bench etc. clear in the introduction to the paper. I believe your contribution is different from theirs since you hone in on a few specific behaviors that models need to be able to replicate to consistently score well in these, and I think you could build a story off that instead of acting like your benchmark is the first to require an LLM to interact sequentially with an environment (which I hope we can agree it is not).
- It would be good if you could simplify the exposition and distill the results into a smaller set of metrics. For example, for the graph exploration tasks your ACC metric seems to capture the notion of a policy metric (as defined by you as "measur[ing] how fast the model's output converges to the final objective") better than the G-sum metric (which I cannot think of a relevant interpretation of).
- I assume it is too late to change your prompts to allow for CoT but I would at least like to see this rather strange design decision justified in the main text, and the relationship to Wei et al. (2022) clarified accordingly.
- Your claim that the embodied environments are harder does not (based on Figure 3) look true to me; on average I would say they look to be about equal. Luckily there's a good way of finding out which one of us is wrong by running some form of hypothesis test on the null hypothesis that the mean accuracies of each setting are equal (eg compute a p-value and present it in the paper to justify this claim)
- I believe you would be able to find prior work discussing the trend you have observed in your ICL study, ie that performance initially drops but then increases. Unfortunately I cannot find the reference, so I might simply be wrong, but if you can find prior work along the lines of "icl first causes memorization and then generalization" then that would strengthen your findings here.
- I think claim (4) in the abstract is a little bit too strong in light of your experiments and the extended version of the claim given in the introduction. I recommend double checking that you feel confident that your experiments support this claim, and if so unifying that with the slightly weaker claim from the introduction.
- I do not think you need the "margin" metrics in your experiments. If the point is to show that the variance (over the task generating distribution) is low, a simple range would have sufficed. This would also reduce the cognitive burden of having to parse yet another new metric, and might free up some space in your tables.
- Typos and spelling mistakes:
  - 2.2, GuessNum: "guess" should be "guessing"
  - 2.2, DFS: "decide" should be "deciding"
  - 3.1: second and fifth sentences of the first paragraph have inconsistent tenses / incorrect sentence structure
  - 4: Incorrect citation format for GPT-4
  - 4.2: "but still not" should be "but are still not"
  - 5: first and second sentence are very difficult to parse

**Strengths And Weaknesses:**

Strengths:
- The paper is mostly well written (a few sections could use some more polishing, but on the whole the presentation is very clear)
- The setting is interesting; although somewhat contrived/synthetic, I think it offers a valuable perspective on a style of reasoning that O1/R1-style CoT RL would *not* immediately help with, since there is no way to know the correct answer without interacting repeatedly with the environment
- The evaluations are thorough; the models considered span several licenses, size ranges and release dates
- The authors do a good job of deriving a few downstream insights from their benchmark (eg the performance gap closing as the number of ICL examples increases)

Weaknesses:
- The biggest weakness to me is that your prompts discourage the mode from doing CoT. I do not understand why you did this; it artificially reduces the amount of computation the model can do (since some computations might require multiple forward
passes), and it makes your framing of Wei et al. (2022) in 3.2 very strange, since you essentially state that it informed your experiment design and then go ahead to set up an experiment that does not leverage the core finding of the paper at all.
- The (in my opinion close) relationship between this benchmark and agent benchmarks is somewhat swept under the rug, being only discussed briefly in the related work when it could have been a solid motivation for why your benchmark (which effectively distills some of the core algorithmic challenges that agents implicitly face in other agent benchmarks) should exist
- There is a lot of bespoke metrics being defined, which makes it very hard to interpret the results, especially as there is already a lot of information to trawl through in the first place due.
- There are a few minor typos here and there.

---

> ### Author Response · Authors · 2025-03-22
>
> We thank the reviewer for the detailed comments and find that our proposed benchmark has an interesting setting offering a important perspective and thorough evaluations, and our observation insightful. We address your concerns as follows:
>
> ## Comparison with agentic benchmarks
> The main difference between our proposed benchmark with existing agentic benchmarks is that we used algorithmic environments, the motivations of which are, stated in Sec. 2.1, 1) scalable dataset size, 2) controllable environment complexity, 3) known optimal policy for easier analysis and providing guidance. We have further clarifed this in the Sec. 5 in our revised paper.
>
> ## Simplified metrics
> In our revised version, we have removed the policy metrics in the tables in Sec. 4 and only shown complete results with policy metrics in Appendix C.
>
> ## Reason for not using CoT
> The main reason we didn't provide results with CoT is that we found it is very hard for models to follow a given format and extract the final answer if we deliberately encourage models to generate extra sentences such as CoT prompting.
>
> In our revised version, we provide the evaluation results of Llama3-70B with CoT on embodied environments in Appendix A, showing that CoT is not consistently helpful across different environments.
>
> ## Other literature discussing the negative effects of ICL
> We have found some literature and added the disccusion about the difference between our conclusion and conclusions from existing literature in the revised Sec. 5.
>
> ## Claim about embodied environments being harder
> After due consideration, we have changed this claim in the revised version. Now it is stated that embodied environments is not consistently harder than base environments even though embodied environments may intuitively require more reasoning ability for models.
>
> ## Reason behind weak models’ under-performance
> As shown by the experimental results with teacher guiding in Sec. 4.5, weak models' performance can be greatly improved by just a few optimal predecessor steps, which indicate that weak models' under-performance is greatly due to the incapability to start well. As it maybe uncertain whether which or the two factors, e.g. the incapability for weak models to start well or maintain performane, contributes more to weak models' failure, we have weakened the claim about this in our revised paper.
>
> ## Necessity of "margin" metrics
> As the "range" is approximately the same as the "margin", we find it necessary to formally define it to avoid ambiguity and clarify how we measure the variance of testing results. Therefore, we decided to keep it but move the detailed discussion to Appendix B, according to your advice.
>
> ## Other writing mistakes
> We have fixed these mistakes in the revised version.

---

> > ### Author Response · Authors · 2025-03-24
> >
> > ## Updated CoT Experiments with Llama3-70B
> > We have updated the evaluation results of Llama3-70B with CoT on base environments in addition to embodied environments in Appendix A. The conclusion drawn from these results is consistent with the previous one that CoT is not consistently helpful across different environments.

---

### Comment · Action_Editor_3Wsq · 2025-06-13
**Action editor requests**

Hi,
Thank you for the camera-ready version and for modifying the paper according to the comments.
However, it seems this request was not taken into account?

`However, I do have a request: Currently there is some discussion on the relation between this benchmark and other benchmarks related to agents and coding. I think the discussion on agent benchmarks should be in the introduction and should mention not only coding benchmarks (swebench, etc.) but also web agent benchmarks (WebArena, Mind2web, and more). The authors should explain in the beginning why do we need their benchmark. The claim that it allows for control and scale is fine but needs to be in the intro not differred to related work, this is a required change.`

Could you please explain or conversely do the required change?

---

> ### Author Response · Authors · 2025-06-15
>
> Dear AE,
>
> We have made such changes in our previous revision. I just uploaded a new camera-ready revision to further clarify this in Sec. 5. Please let us know if there are any more problems.
>
> Best,
> Authors

---

> > ### Comment · Action_Editor_3Wsq · 2025-06-16
> > **Not quite there yet**
> >
> > As I explained in my comment, I think by section 5 it is too late, and this requires a change to the introduction.
> >
> > My explicit request is to add a sentence at the end of the second paragraph saying that there are benchmarks for agents that address some of the limitations described  (e.g., they are not one-off etc.), they still have issues as explained in section 5 and have proper citations. Then the third paragraph can talk about your benchmark bridging this gap. This seems like a more fair characterization.
> >
> > Also in section 5 now there is a type, you are missing a space after "e.g."
> >
> > Please let's try to perform the requested changes as they are described to not have too much back and forth I don't find my request to be so unusual. Feel free to express if you feel this request for 1-2 additional sentences in the second paragraph is problematic.

---

> > > ### Author Response · Authors · 2025-06-17
> > > **Updated the Introduction and fixed typos in Sec. 5**
> > >
> > > Dear AE,
> > >
> > > We are sorry for the earlier confusion about your request. We have just updated the Introduction with the discussion about the comparison with existing agentic benchmarks and fixed typos in Sec. 5. We have marked the changes with purple fonts, which will be changed to normal in the final camera-ready revision.
> > >
> > > Please let us know whether this revision addresses your request.
> > >
> > > Thanks,
> > > Authors

---

> > > > ### Comment · Action_Editor_3Wsq · 2025-06-19
> > > > **Looks good to me.**
> > > >
> > > > The placing of figure 1 now is not super pertty consider changing it a bit.

---

### Comment · Editors_In_Chief · 2025-07-01

The authors submitted a camera-ready revision on June 17, 2025 which was approved by the Action Editor on June 18, 2025. Unfortunately, the approved version was missing the publication dates in the header, and highlighted some changes in color. On July 1, 2025, the EiCs uploaded a new version of the paper which fixes both these issues, includes a Github link, fixed a typo in Appendix C.1, and added model checkpoint information in Appendix E.

---

### Decision · Action_Editor_3Wsq · 2025-04-14

**Recommendation:** Accept with minor revision

**Comment:**

The primary contribution of this work in the provided benchmark itself. Reviewers seem to agree that the existence of this benchmark will be useful for people working on agents and reasoning in environments.

However, I do have a request:
Currently there is some discussion on the relation between this benchmark and other benchmarks related to agents and coding. I think the discussion on agent benchmarks should be in the introduction and should mention not only coding benchmarks (swebench, etc.) but also web agent benchmarks (WebArena, Mind2web, and more). The authors should explain in the beginning why do we need their benchmark. The claim that it allows for control and scale is fine but needs to be in the intro not differred to related work, this is a required change.

The second contribution is the evaluation of models on this benchmark. On this front, most of the models evaluated are what is often thought of as "old" nowadays. I was glad to see that the authors added llama-3-70b which is newer (came out april 2024). However, there is no discussion in the text about the results of llama-3-70b even though it does well. For example in 4.2 it is discussed that open models are much worse than closed models but this is not accurate in terms of this newer model added. The authors must change the text to reflect the results of the strongest open source model they present. This is a required change.

Moreover, since December 2024 there has been a lot of interest in reasoning models (o1, gemini-thinking, deepseek-r1, etc.).
This is a benchmark on reasoning so if the authors actually want people to appreciate the empirical work done it seems necessary to evaluate reasoning models at this point. Seems possible to do this for Deepseek-r1 distilled model, there are 8b/70b based on llama and other sizes based on qwen. I know this might not be seen as a "minor revision" but I view this as important for a benchmark that is focused on reasoning. It should be easy given that weights are open.

Last, I would recommend in general to evaluate newer models. Llama-2 is considered quite old. If authors want researchers to read the paper it would serve them to use modern models. Inevitably benchmarks saturate so evaluating with the right models can influence how much the benchmark will be used down the line. This is **not** required by me, only a thought the authors can consider.

A few more asks:
* please proofread citations and use \citet and \citep appropriately so that citations that are not part of the text are in parenthesis and citations that are part of the text are presented with only the year in parenthesis.
* "This contradicts common assertions in LLM development and points to an oversight of sequential reasoning capabilities and overfitting during in-context learning in current LLM research". The part after "overfitting..." is unclear in my opinion it would be good to clarify (not required but advised)








A few more minor requests:

**Audience:**

This is a benchmark on the reasoning abilities of LLMs in a simulated agentic setting - this is perhaps one of the most popular topics nowadays in the ML community.

**Claims And Evidence:**

This work presents a benchmark that checks the ability of LLMs to handle sequential reasoning tasks in an algorithmic environment.
They then evaluate a large set of models in easy/hard base/embodied environments also controlling for in-context examples. They show some interesting empirical findings regarding the importance of scale and in-context examples.

Two things that are currently missing and I will discuss below:
a. More front and center discussion of agentic tasks that exist and the relation of this work to them.
b. More emphasis in the text on more modern models (llama3-70b, see below)

---

> ### Author Response · Authors · 2025-06-09
>
> Dear AE,
>
> We are deeply grateful for the efforts of all the editors and reviewers. Your insightful feedback has significantly enhanced the quality of our paper. To highlight the key revisions in our camera-ready submission, we have summarized the main changes below:
>
> 1. In Sec. 4.2, we added more discussion about Llama30-70B-Instruct and included results from a reasoning model, e.g. o1-preview.
> 2. In Sec. 5, we corrected the usage of \citep and \citet.
> 3. In Sec. 1, we clarified the sentence "This contradicts common assertions in LLM development and points to an oversight ..."
>
> Thanks,
> Authors